# DrS: Learning Reusable Dense Rewards for Multi-Stage Tasks

**Tongzhou Mu, Minghua Liu, Hao Su**
UC San Diego
{t3mu,mil070,haosu}@ucsd.edu

## Abstract

The success of many RL techniques heavily relies on human-engineered dense rewards, which typically demands substantial domain expertise and extensive trial and error. In our work, we propose **DrS** (**D**ense **r**eward learning from **S**tages), a novel approach for learning *reusable* dense rewards for multi-stage tasks in a data-driven manner. By leveraging the stage structures of the task, DrS learns a high-quality dense reward from sparse rewards and demonstrations if given. The learned rewards can be *reused* in unseen tasks, thus reducing the human effort for reward engineering. Extensive experiments on three physical robot manipulation task families with 1000+ task variants demonstrate that our learned rewards can be reused in unseen tasks, resulting in improved performance and sample efficiency of RL algorithms. The learned rewards even achieve comparable performance to human-engineered rewards on some tasks. See our project page for more details.

## 1 Introduction

The success of many reinforcement learning (RL) techniques heavily relies on dense reward functions (Hwangbo et al., 2019; Peng et al., 2018), which are often tricky to design by humans due to heavy domain expertise requirements and tedious trials and errors. In contrast, sparse rewards, such as a binary task completion signal, are significantly easier to obtain (often directly from the environment). For instance, in pick-and-place tasks, the sparse reward could simply be defined as the object being placed at the goal location. Nonetheless, sparse rewards also introduce challenges (e.g., exploration) for RL algorithms (Pathak et al., 2017; Burda et al., 2018; Ecoffet et al., 2019). Therefore, a crucial question arises: *can we learn dense reward functions in a data-driven manner?*

Ideally, the learned reward will be **reused** to efficiently solve new tasks that share similar success conditions with the task used to learn the reward. For example, in pick-and-place tasks, different objects may need to be manipulated with varying dynamics, action spaces, and even robot morphologies. For clarity, we refer to each variant as a *task* and the set of all possible pick-and-place tasks as a *task family*. Importantly, the reward function, which captures approaching, grasping, and moving the object toward the goal position, can potentially be transferred within this task family. This observation motivates us to explore the concept of *reusable rewards*, which can be learned as a function from some tasks and reused in unseen tasks. While existing literature in RL primarily focuses on the reusability (generalizability) of policies, we argue that rewards can pose greater flexibility for reuse across tasks. For example, it is nearly impossible to directly transfer a policy operating a two-finger gripper for pick-and-place to a three-finger gripper due to action space misalignment, but a reward inducing the approach-grasp-move workflow may apply for both types of grippers.

However, many existing works on reward learning do not emphasize reward reuse for new tasks. The field of learning a reward function from demonstrations is known as inverse RL in the literature (Ng et al., 2000; Abbeel & Ng, 2004; Ziebart et al., 2008). More recently, adversarial imitation learning (AIL) approaches have been proposed (Ho & Ermon, 2016; Kostrikov et al., 2018; Fu et al., 2017; Ghasemipour et al., 2020) and gained popularity. Following the paradigm of GANs (Goodfellow et al., 2020), AIL approaches employ a policy network to generate trajectories and train a discriminator to distinguish between agent trajectories from demonstration ones. By using the discriminator score as rewards, (Ho & Ermon, 2016) shows that a policy can be trained to imitate the demonstrations. Unfortunately, such rewards are *not reusable* across tasks – at convergence, the discriminator outputs

$\frac{1}{2}$ for both the agent trajectories and the demonstrations, as discussed in (Goodfellow et al., 2020; Fu et al., 2017), making it unable to learn useful information for solving new tasks.

In contrast to AIL, we propose a novel approach for learning *reusable rewards*. Our approach involves incorporating sparse rewards as a supervision signal in lieu of the original signal used for classifying demonstration and agent trajectories. Specifically, we train a discriminator to classify *success trajectories* and *failure trajectories* based on the binary sparse reward. Please refer to Fig. 2 (a)(b) for an illustrative depiction. Our formulation assigns higher rewards to transitions in success trajectories and lower rewards to transitions within failure trajectories, which is consistent throughout the entire training process. As a result, the reward will be reusable once the training is completed. Expert demonstrations can be included as success trajectories in our approach, though they are not mandatory. We only require the availability of a sparse reward, which is a relatively weak requirement as it is often an inherent component of the task definition.

Our approach can be extended to leverage the inherent structure of *multi-stage tasks* and derive stronger dense rewards. Many tasks naturally exhibit multi-stage structures, and it is relatively easy to assign a binary indicator on whether the agent has entered a stage. For example, in the "Open Cabinet Door" task depicted in Fig. 1, there are three stages: 1) approach the door handle, 2) grasp the handle and pull the door, and 3) release the handle and keeping it steady. If the agent is grasping the handle of the door but the door has not been opened enough, then we can simply use a corresponding binary indicator asserting that the agent is in the 2nd stage. [1] By utilizing these stage indicators, we can learn a dense reward for each stage and combine them into a more structured reward. Since the horizon for each stage is shorter than that of the entire task, learning a high-quality dense reward becomes more feasible. Furthermore, this approach provides flexibility in incorporating extra information beyond the final success signal. We dub our approach as **DrS** (**D**ense **r**eward learning from **S**tages).

Our approach exhibits strong performance on challenging tasks. To assess the reusability of the rewards learned by our approach, we employ the ManiSkill benchmark (Mu et al., 2021; Gu et al., 2023), which offers a large number of task variants within each task family.We evaluate our approach on three task families: Pick-and-Place, Open Cabinet Door, and Turn Faucet, including 1000+ task variants. Each task variant involves manipulating a different object and requires precise low-level physical control, thereby highlighting the need for a good dense reward. Our results demonstrate that the learned rewards can be reused across tasks, leading to improved performance and sample efficiency of RL algorithms compared to using sparse rewards. In certain tasks, the learned rewards even achieve performance comparable to those attained by human-engineered reward functions.

Moreover, our approach **drastically reduces the human effort** needed for reward engineering. For instance, while the human-engineered reward for "Open Cabinet Door" involves *over 100 lines of code, 10 candidate terms, and tons of "magic" parameters*, our approach only requires *two boolean functions* as stage indicators: if the robot has grasped the handle and if the door is open enough. See appendix B for a detailed example illustrating how our method reduces the required human effort.

Our contributions can be summarized as follows:

- We propose **DrS** (**D**ense **r**eward learning from **S**tages), a novel approach for learning reusable dense rewards for multi-stage tasks, effectively reducing human efforts in reward engineering.
- Extensive experiments on 1,000+ task variants from three task families showcase the effectiveness of our approach in generating high-quality and reusable dense rewards.

## 2 RELATED WORKS

**Learning Reward from Demo (Offline)** Designing rewards is challenging due to domain knowledge requirements, so approaches to learning rewards from data have gained attention. Some methods adopt classification-based rewards, i.e., training a reward by classifying goals (Smith et al., 2019; Kalashnikov et al., 2021; Du et al., 2023) or demonstration trajectories (Zolna et al., 2020). Other methods (Zakka et al., 2022; Aytar et al., 2018) use the distance to goal as a reward function, where the distance is usually computed in a learned embedding space, but these methods usually require that the goal never changes in a task. These rewards are only trained on offline datasets, hence they can *easily be exploited* by an RL agent, i.e., an RL can enter a state that is not in the dataset and get a wrong reward signal, as studied in (Vecerik et al., 2019; Xu & Denil, 2021).

---

[1]Stage indicators are only required during RL training, but **not required** when deploying policy to real world.

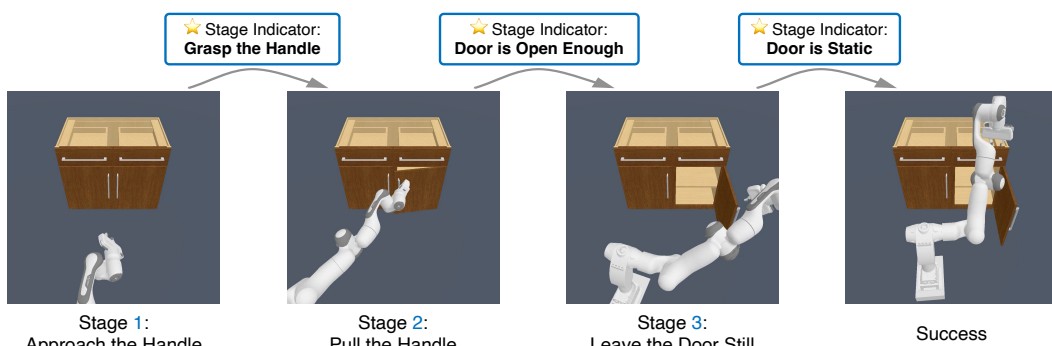

Figure 1: An illustration of stage indicators in an OpenCabinetDoor task, which can be naturally divided into three stages plus a success state. A stage indicator is a binary function representing whether the current state is in a certain stage, and it can be simply defined by some boolean functions.

**Learning Reward from Demo (Online)** The above issue can be addressed by allowing agents to verify the reward in the environment, and inverse reinforcement learning (IRL) is the prominent paradigm. IRL aims to recover a reward function given expert demonstrations. Traditional IRL methods (Ng et al., 2000; Abbeel & Ng, 2004; Ziebart et al., 2008; Ratliff et al., 2006) often require multiple iterations of Markov Decision Process solvers (Puterman, 2014), resulting in poor sample efficiency. In recent years, adversarial imitation learning (AIL) approaches are proposed (Ho & Ermon, 2016; Kostrikov et al., 2018; Fu et al., 2017; Ghasemipour et al., 2020; Liu et al., 2019). They operate similarly to generative adversarial networks (GANs) (Goodfellow et al., 2020), in which a generator (the policy) is trained to maximize the confusion of a discriminator, and the discriminator (serves the role of rewards) is trained to classify the agent trajectories and demonstrations. However, such rewards are *not reusable* as we discussed in the introduction - classifying agent trajectories and demonstrations is impossible at convergence. In contrast, our approach gets rid of this issue by classifying the success/failure trajectories instead of expert/agent trajectories.

**Learning Reward from Human Feedback** Recent studies (Christiano et al., 2017; Ibarz et al., 2018; Jain et al., 2013) infer the reward through human preference queries on trajectories or explicitly asking for trajectory rankings (Brown et al., 2019). Another line of works (Fu et al., 2018; Singh et al., 2019) involves humans specifying desired outcomes or goals to learn rewards. However, in these methods, the rewards only distinguish goal from non-goal states, offering relatively weak incentives to agents at the beginning of an episode, especially in long-horizon tasks. In contrast, our approach classifies all the states in the trajectories, providing strong guidance throughout the entire episode.

**Reward Shaping** Reward shaping methods aim to densify sparse rewards. Earlier works (Ng et al., 1999) study the forms of shaped rewards that induce the same optimal policy as the ground-truth reward. Recently, some works (Trott et al., 2019; Wu et al., 2021) have shaped the rewards as the distance to the goal, similar to some offline reward learning methods mentioned above. Another idea (Memarian et al., 2021) involves shaping delayed reward by ranking trajectories based on a fine-grained preference oracle. In contrast to these reward shaping approaches, our method leverages demonstrations, which are available in many real-world problems (Sun et al., 2020; Dasari et al., 2019). This not only boosts the reward learning process but also reduces the additional domain knowledge required by these methods.

**Task Decomposition** The decomposition of tasks into stages/sub-tasks has been explored in various domains. Hierarchical RL approaches (Frans et al., 2018; Nachum et al., 2018; Levy et al., 2018) break down policies into sub-policies to solve specific sub-tasks. Skill chaining methods (Lee et al., 2021; Gu et al., 2022; Lee et al., 2019) focus on solving long-horizon tasks by combining multiple short-horizon policies or skills. Recently, language models have also been utilized to break the whole task into sub-tasks Ahn et al. (2022). In contrast to these approaches that utilize stage structures in policy space, our work explores an orthogonal direction by designing rewards with stage structures.

## 3 PROBLEM SETUP

In this work, we adopt the Markov Decision Process (MDP) $\mathcal{M} := \langle S, A, T, R, \gamma \rangle$ as the theoretical framework, where $R$ is a reward function that defines the goal or purpose of a task. Specifically, we focus on tasks with *sparse rewards*. In this context, "sparse reward" denotes a binary reward function that gives a value of 1 upon successful task completion and 0 otherwise:

Figure 2: a) GAIL's discriminator aims to distinguish agent trajectories from demonstrations. b) In single-stage tasks, the discriminator in our approach aims to distinguish success trajectories from failure ones. c) In multi-stage tasks, our approach train a separate discriminator for each stage. The discriminator for stage $k$ aims to distinguish trajectories that reach beyond stage $k$ from those that only reach up to stage $k$. d) Overall, our approach has 2 phases: **reward learning** and **reward reuse**.

$$R_{sparse}(s) = \begin{cases} 1 & \text{task is completed by reaching one of the success states } s \\ 0 & \text{otherwise} \end{cases} \quad (1)$$

Our objective is to learn a dense reward function from a set of training tasks, with the intention of reusing it for unseen test tasks. Specifically, we aim to successfully train RL agents from scratch on the test tasks using the learned rewards. The desired outcome is to enhance the efficiency of RL training, surpassing the performance achieved by sparse rewards.

We assume that both the training and test tasks are in the same *task family*. A task family refers to a set of task variants that share the same success criteria, but may differ in terms of assets, initial states, transition functions, and other factors. For instance, the task family of object grasping includes tasks such as "Alice robot grasps an apple" and "Bob robot grasps a pen." The key point is that tasks within the same task family share a common underlying sparse reward.

Additionally, we posit that the task can be segmented into multiple stages, and the agent has access to several *stage indicators* obtained from the environment. A stage indicator is a binary function that indicates whether the current state corresponds to a specific stage of the task. An example of stage indicators is in Fig. 1. This assumption is quite general as many long-term tasks have multi-stage structures, and determining the current stage of the task is not hard in many cases. By utilizing these stage indicators, it becomes possible to construct a reward that is slightly denser than the binary sparse reward, which we refer to as a *semi-sparse* reward, and it serves as a strong baseline:

$$R_{semi-sparse}(s) = k, \text{ when state } s \text{ is at stage } k \quad (2)$$

We aim to design an approach that learns a dense reward based on the stage indicators. When expert demonstration trajectories are available, they can also be incorporated to boost the learning process.

Note that the stage indicators are only required during RL training, but *not required when deploying the policy to the real world*. Training RL agents directly in the real world is often impractical due to cost and safety issues. Instead, a more common practice is to train the agent in simulators and then transfer/deploy it to the real world. While obtaining the stage indicators in simulators is fairly easy, it is also possible to obtain them in the real world by various techniques (robot proprioception, tactile sensors Lin et al. (2022); Melnik et al. (2021), visual detection/tracking Kalashnikov et al. (2018; 2021), large vision-language models Du et al. (2023), etc.).

## 4 DRS: DENSE REWARD LEARNING FROM STAGES

Dense rewards are often tricky to design by humans (see an example in appendix B), so we aim to learn a reusable dense reward function from stage indicators in multi-stage tasks and demonstrations when available. Overall, our approach has two phases, as shown in Fig. 2 (d):

- **Reward Learning Phase**: learn the dense reward function using training tasks.
- **Reward Reuse Phase**: reuse the learned dense reward to train new RL agents in test tasks.

Since the reward reuse phase is just a regular RL training process, we only discuss the reward learning phase in this section. We first explain how our approach learns a dense reward in one-stage tasks (Sec. 4.1). Then, we extend this approach to multi-stage tasks (Sec. 4.2).

### 4.1 REWARD LEARNING ON ONE-STAGE TASKS

In line with previous work (Vecerik et al., 2019; Fu et al., 2018), we employ a classification-based dense reward. We train a classifier to distinguish between good and bad trajectories, utilizing the learned classifier as dense reward. Essentially, states resembling those in good trajectories receive higher rewards, while states resembling bad trajectories receive lower rewards. While previous Adversarial Imitation Learning (AIL) methods (Ho & Ermon, 2016; Kostrikov et al., 2018) used discriminators as classifiers/rewards to distinguish between agent and demonstration trajectories, these discriminators cannot be directly *reused* as rewards to train new RL agents. As the policy improves, the agent trajectories (negative data) and the demonstrations (positive data) can become nearly identical. Therefore, at convergence, the discriminator output for both agent trajectories and demonstrations tends to approach $\frac{1}{2}$, as observed in GANs (Goodfellow et al., 2020) (also noted by (Fu et al., 2017; Xu & Denil, 2021)). This makes it unable to learn useful info for solving new tasks.

Our approach introduces a simple modification to existing AIL methods to ensure that the discriminator continues to learn meaningful information even at convergence. The key issue previously mentioned arises from the diminishing gap between agent and demonstration trajectories over time, making it challenging to differentiate between positive and negative data. To address this, we propose training the discriminator to distinguish between success and failure trajectories instead of agent and demonstration trajectories. By defining success and failure trajectories based on the sparse reward signal from the environment, the gap between them remains intact and does not shrink. Consequently, the discriminator effectively emulates the sparse reward signal, providing dense reward signals to the RL agent. Intuitively, a state that is closer to the success states in terms of task progress (rather than Euclidean distance) receives a higher reward, as it is more likely to occur in success trajectories. Fig. 2(a) and (b) illustrate the distinction between our approach and traditional AIL methods.

To ensure that the training data consistently includes both success and failure trajectories, we use replay buffers to store historical experiences, and train the discriminator in an off-policy manner. While the original GAIL is on-policy, recent AIL methods (Kostrikov et al., 2018; Orsini et al., 2021) have adopted off-policy training for better sample efficiency. Note that although our approach shares similarities with AIL methods, it is not adversarial in nature. In particular, our policy does not aim to deceive the discriminator, and the discriminator does not seek to penalize the agent's trajectories.

### 4.2 REWARD LEARNING ON MULTI-STAGE TASKS

In multi-stage tasks, it is desirable for the reward of a state in stage $k + 1$ to be strictly higher than that of stage $k$ to incentivize the agent to progress towards later stages. The semi-sparse reward (Eq. 2) aligns with this intuition, but it is still a bit too sparse. If each stage of the task is viewed as an individual task, the semi-sparse reward acts as a sparse reward for each stage. In the case of a one-stage task, a discriminator can be employed to provide a dense reward. Similarly, for multi-stage tasks, a separate discriminator can be trained for each stage to serve as a dense reward for that particular stage. By training stage-specific discriminators, we can effectively address the sparse reward issue and guide the agent's progress through the different stages of the

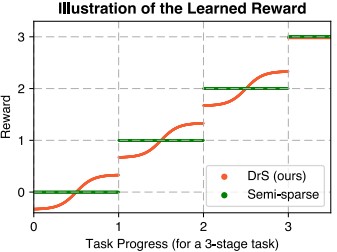

Figure 3: An illustration of our learned reward, which fills the gaps in semi-sparse rewards, resulting in a smooth reward curve.

task. Fig. 3 gives an intuitive illustration of our learned reward, which fills the gaps in semi-sparse rewards, resulting in a smooth reward curve.

To train the discriminators for different stages, we need to establish the positive and negative data for each discriminator. In one-stage tasks, positive data comprises success trajectories and negative data encompasses failure trajectories. In multi-stage tasks, we adopt a similar approach with a slight modification. Specifically, we assign a stage index to each trajectory, which is determined as the highest stage index among all states within the trajectory:

$$\text{StageIndex}(\tau : (s_0, s_1, ...)) = \max_i \text{StageIndex}(s_i), \qquad (3)$$

where $\tau$ is a trajectory and $s_i$ are the states in $\tau$. For the discriminator associated with stage $k$, positive data consists of trajectories that progress beyond stage $k$ (StageIndex $> k$), and negative data consists of trajectories that reach up to stage $k$ (StageIndex $\leq k$).

Once the positive and negative data for each discriminator have been established, the next step is to combine these discriminators to create a reward function. While the semi-sparse reward (Eq. 2) lacks incentives for the agent at stage $k$ until it reaches stage $k + 1$, we can fill in the gaps in the semi-sparse reward by the stage-specific discriminators. We define our learned reward function for a multi-stage task as follows:

$$R(s') = k + \alpha \cdot \tanh(\text{Discriminator}_k(s')) \tag{4}$$

where $k$ is the stage index of $s'$ and $\alpha$ is a hyperparameter. Basically, the formula incorporates a dense reward term into the semi-sparse reward. The $\tanh$ function is used to bound the output of the discriminators. As the range of the $\tanh$ function is (-1, 1), any $\alpha < \frac{1}{2}$ ensures that the reward of a state in stage $k + 1$ is always higher than that of stage $k$. In practice, we use $\alpha = \frac{1}{3}$ and it works well.

### 4.3 IMPLEMENTATION

From the implementation perspective, our approach is similar to GAIL, but with a different training process for discriminators. While the original GAIL is combined with TRPO (Schulman et al., 2015), (Orsini et al., 2021) found that using state-of-the-art off-policy RL algorithms (like SAC (Haarnoja et al., 2018) or TD3 (Fujimoto et al., 2018)) can greatly improve the sample efficiency of GAIL. Therefore, we also combine our approach with SAC, and the full algorithm is summarized in Algo. 1.

In addition to the regular replay buffer used in SAC, our approach maintains $N$ different stage buffers to store trajectories corresponding to different stages(defined by Eq. 3). Each trajectory is assigned to only one stage

---

**Algorithm 1 DrS (Dense reward learning from Stages)**

**Require:** Task MDP $\mathcal{M}$, Number of stages in task $N$, Demonstration dataset $\mathcal{D} := \{\tau^0, \tau^1, ...\}$ (optional)
1: Initialize policy $\pi$, critic $Q$, replay buffer $\mathcal{B}_R$
2: Initialize discriminators $f_0, f_1, ..., f_{N-1}$, stage buffers $\mathcal{B}_0, \mathcal{B}_2, .., \mathcal{B}_N$
3: Fill demo $\mathcal{D}$ into $\mathcal{B}_\mathcal{N}$: $\mathcal{B}_\mathcal{N} \leftarrow \mathcal{B}_\mathcal{N} \cup \mathcal{D}$
4: **for** each iteration **do**
5:     Collect trajectories $\{\tau^0_\pi, \tau^1_\pi, ...\}$ by executing $\pi$ in $\mathcal{M}$
6:     Add trajectories to replay buffer: $\mathcal{B}_R \leftarrow \mathcal{B}_R \cup \{\tau^0_\pi, \tau^1_\pi, ...\}$
7:     **for** each trajectory $\tau^i_\pi$ in $\{\tau^0_\pi, \tau^1_\pi, ...\}$ **do**
8:         $j = \text{StageIndex}(\tau^i_\pi)$ according to Eq. 3
9:         $\mathcal{B}_j \leftarrow \mathcal{B}_j \cup \{\tau^i_\pi\}$
10:     **for** each gradient step for discriminators **do**
11:         **for** each discriminator $f_k$ **do**
12:             Sample negative data from $\bigcup_{i=0}^{k} \mathcal{B}_i$
13:             Sample positive data from $\bigcup_{i=k+1}^{N} \mathcal{B}_i$
14:             Update $f_k$ using BCE loss
15:     **for** each gradient step for the policy $\pi$ **do**
16:         Sample from $\mathcal{B}_R$
17:         Compute rewards according to Eq. 4
18:         Update $\pi$ and $Q$ by SAC (Haarnoja et al., 2018)

---

buffer based on its stage index. During the training of the discriminators, we sample data from the union of multiple buffers. In practice, we early stop the discriminator training of $k$ once its success rate is sufficiently high, as we find it reduces the computational cost and makes the learned reward more robust. Note that our approach uses the next state $s'$ as the input to the reward, which aligns with common practices in human reward engineering (Gu et al., 2023; Zhu et al., 2020). However, our approach is also compatible with alternative forms of input, such as $(s, a)$ or $(s, a, s')$.

## 5 EXPERIMENTS

### 5.1 SETUP AND TASK DESCRIPTIONS

We evaluated our approach on three challenging physical manipulation task families from the ManiSkill (Mu et al., 2021; Gu et al., 2023): Pick-and-Place, Turn Faucet, and Open Cabinet Door. Each task family includes a set of different objects to be manipulated. To assess the reusability of the learned rewards, we divided the objects within each task family into non-overlapping training and test sets, as depicted in Fig.4. During the reward learning phase, we learned the rewards by training an agent for each task family to manipulate all training objects. In the subsequent reward reuse phase, the learned reward rewards are reused to train an agent to manipulate all test objects for each task family. And we compare with other baseline rewards in this **reward reuse** phase. It is important to note that our learned rewards are agnostic to the specific RL algorithm employed. However, we utilized the Soft Actor-Critic (SAC) algorithm to evaluate the quality of the different rewards.

To assess the *reusability* of the learned rewards, it is crucial to have a diverse set of tasks that exhibit similar structures and goals but possess variations in other aspects. However, most existing

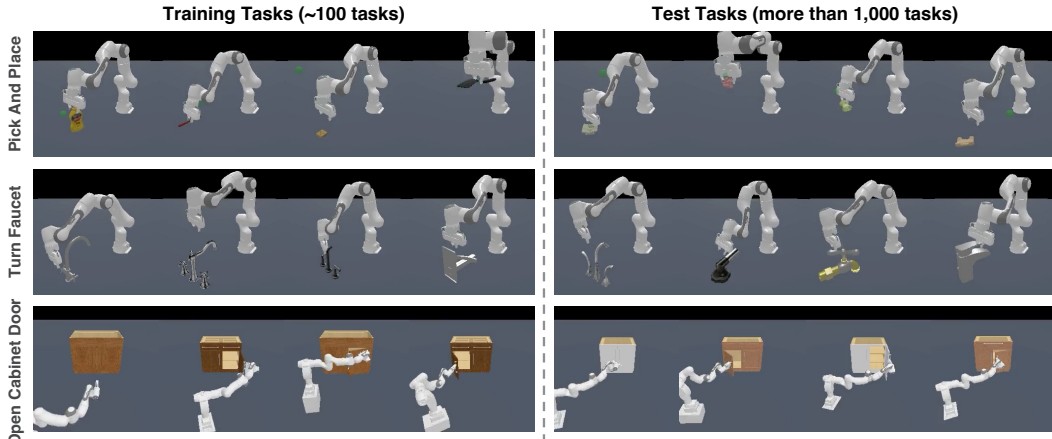

Figure 4: We evaluated our approach DrS on more than 1,000 task variants from three task families in ManiSkill (Mu et al., 2021; Gu et al., 2023). Each task variant is associated with a different object. All tasks require low-level physical control. The objects in training and test tasks are non-overlapped.

benchmarks lack an adequate number of task variations within the same task family. As a result, we primarily conducted our evaluation on the ManiSkill benchmark, which offers a range of object variations within each task family. This allowed us to thoroughly evaluate our learned rewards in a realistic and comprehensive manner.

**Pick-and-Place:** A robot arm is tasked with picking up an object and relocating it to a random goal position in mid-air. The task is completed if the object is in close proximity to the goal position, and both the robot arm and the object remain stationary. The stage indicators include: (a) the gripper grasps the object, (b) the object is close the goal position, and (c) both the robot and the object are stationary. We learn rewards on 74 YCB objects and reuse rewards on 1,600 EGAD objects.

**Turn Faucet:** A robot arm is tasked to turn on a faucet by rotating its handle. The task is completed if the handle reaches a target angle. The stage indicators include: (a) the target handle starts moving, (b) the handle reaches a target angle. We learn rewards on 10 faucets and reuse rewards on 50 faucets.

**Open Cabinet Door:** A single-arm mobile robot is required to open a designated target door on a cabinet. The task is completed if the target door is opened to a sufficient degree and remains stationary. The stage indicators include: (a) the robot grasps the door handle, (b) the door is open enough, and (c) the door is stationary. We learn rewards on 4 cabinet doors and reuse rewards on 6 cabinet doors. Note that we remove all single-door cabinets in this task family, as they can be solved by kicking the side of the door and this behavior can be readily learned by sparse rewards.

We employed low-level physical control for all task families. Please refer to the appendix A for a detailed description of the object sets, action space, state space, and demonstration trajectories.

## 5.2 BASELINES

**Human-Engineered** The original human-written dense rewards in the benchmark, which require a significant amount of domain knowledge, thus can be considered as an *upper bound* of performance.

**Semi-Sparse** The rewards constructed based on the stage indicators, as discussed in Eq. 2. The agent receives a reward of $k$ when it is in stage $k$. This baseline extends the binary sparse reward.

**VICE-RAQ (Singh et al., 2019)** An improved version of VICE (Fu et al., 2018). It learns a classifier, where the positive samples are successful states annotated by querying humans, and the negative samples are all other states collected by the agent. Since our experiments do not involve human feedback, we let VICE-RAQ query the oracle success condition infinitely for a fair comparison.

**ORIL (Zolna et al., 2020)** A representative offline reward learning method, where the agent does not interact with the environments but purely learns from the demonstrations. It learns a classifier (reward) to distinguish between the states from success trajectories and random trajectories.

## 5.3 COMPARISON WITH BASELINE REWARDS

We trained RL agents using various rewards and assessed the reward quality based on both the sample efficiency and final performance of the agents. The experimental results, depicted in Fig. 5,

Figure 5: Evaluation results of **reusing learned rewards**. All curves use SAC to train, but with different rewards. VICE-RAQ and ORIL get no success. 5 random seeds, the shaded region is std.

demonstrate that our learned reward surpasses semi-sparse rewards and all other reward learning methods across all three task families. This outcome suggests that our approach successfully acquires high-quality rewards that significantly enhance RL training. Remarkably, our learned rewards even achieve performance comparable to human-engineered rewards in Pick-and-Place and Turn Faucet.

Semi-sparse rewards yielded limited success within the allocated training budget, suggesting that RL agents face exploration challenges when confronted with sparse reward signals. VICE-RAQ failed in all tasks. Notably, it actually failed during the reward learning phase on the training tasks, rendering the learned rewards inadequate for supporting RL training on the test tasks. This failure aligns with observations made by (Wu et al., 2021). We hypothesize that by only classifying the success states from other states, it can provide sufficient guidance during the early stages of training, where most states are distant from the success states and receive low rewards. Unsurprisingly, ORIL does not get any success on all tasks either. Without interacting with the environments to gather more data, the learned reward functions easily tend to overfit the provided dataset. When using such rewards in RL, the flaws in the learned rewards are easily exploited by the RL agents.

## 5.4 ABLATION STUDY

We examined various design choices within our approach on the Pick-and-Place task family.

### 5.4.1 ROBUSTNESS TO STAGE CONFIGURATIONS

Though many tasks present a natural structure of stages, there are still different ways to divide a task into stages. To assess the robustness of our approach in handling different task structures, we experiment with different numbers of stages and different ways to define stage indicators.

**Number of Stages** The Pick-and-Place task family originally consisted of three stages: (a) approach the object, (b) move the object to the goal, and (c) make everything stationary. We explored two ways of reducing the number of stages to two, namely merging stages (a) and (b) or merging stages (b) and (c), as well as the 1-stage case. Our results, presented in Fig. 6, indicate that the learned rewards with 2 stages can still effectively train RL agents in test tasks, albeit with lower sample efficiency than those with 3 stages. Specifically, the reward that preserves stage (c) "make everything stationary" performs slightly better than the reward that preserves stage (a) "approach the object". This suggests that it may be more challenging for a robot to learn to stop abruptly without a dedicated stage. However, when reducing the number of stages to 1, the learned reward failed to train RL agents in test tasks, demonstrating the benefit of using more stages in our approach.

**Definition of Stages** The stage indicator "object is placed" is initially defined as if the distance between the object and the goal is less than 2.5 cm. We create two variants of it, where the distance thresholds are 5cm and 10cm, respectively. The results, as depicted in Fig. 7, demonstrate that changing the distance threshold within a reasonable range does not significantly affect the efficiency of RL training. Note that the task success condition is unchanged, and our rewards consistently encourage the agents to reach the success state as it yields the highest reward according to Eq. 4. The stage definitions solely affect the efficiency of RL training during the reward reuse phase.

Overall, the above results highlight the robustness of our approach to different stage configurations, indicating that it is not heavily reliant on intricate stage designs. This robustness contributes to a significant reduction in the burden of human reward engineering.

Figure 6: Ablation study on the number of stages, see here.

Figure 7: Ablation study on the stage definitions, see here.

Figure 8: Fine-tune the policy from reward learning, see here.

### 5.4.2 FINE-TUNING POLICY

In our previous experiments, we assessed the quality of the learned reward by reusing it in training RL agents from scratch since it is the most common and natural way to use a reward. However, our approach also produces a policy as a byproduct in the reward learning phase. This policy can also be fine-tuned using various rewards in new tasks, providing an alternative to training RL agents from scratch. We compare the fine-tuning of the byproduct policy using human-engineered rewards, semi-sparse rewards, and our learned rewards.

As shown in Fig. 8, all policies improve rapidly at the beginning due to the good initialization of the policies. However, fine-tuning with our learned reward yields the best performance (even slightly better than the human-engineered reward), indicating the advantages of utilizing our learned dense reward even with a good initialization. Furthermore, the significant variance observed when fine-tuning the policy with semi-sparse rewards highlights the limitations of sparse reward signals in effectively training RL agents, even with a very good initialization.

### 5.4.3 ADDITIONAL ABLATION STUDIES

Additional ablation studies are provided in appendix E, with key conclusions summarized as follows:

- DrS is compatible with various modalities of reward input, including point cloud data. E.1
- Reward learned by GAIL, even with stage indicators, is not reusable. E.2
- The way of combining the dense rewards from each stage matters. E.3

## 6 CONCLUSION AND LIMITATIONS

To make RL a more widely applicable tool, we have developed a data-driven approach for learning dense reward functions that can be reused in new tasks from sparse rewards. We have evaluated the effectiveness of our approach on robotic manipulation tasks, which have high-dimensional action spaces and require dense rewards. Our results indicate that the learned dense rewards are effective in transferring across tasks with significant variation in object geometry. By simplifying the reward design process, our approach paves the way for scaling up RL in diverse scenarios.

We would like to discuss two main limitations when using the multi-stage version of our approach.

Firstly, though our experiments show the substantial benefits of knowing the multi-stage structure of tasks (at training time, not needed at policy deployment time), we did not specifically investigate how this knowledge can be acquired. Much future work on be done here, by leveraging large language models such as ChatGPT (OpenAI, 2023) (by our testing, they suggest stages highly aligned to the ones we adopt by intuition for all tasks in this work) or employing information-theoretic approaches. Further discussions regarding this point can be found in appendix F.

Secondly, the reliance on stage indicators adds a level of inconvenience when directly training RL agents in the real world. While it is infrequent to directly train RL agents in the real world due to cost and safety issues, when necessary, stage information can still be obtained using existing techniques, similar to (Kalashnikov et al., 2018; 2021). For example, the "object is grasped" indicator can be acquired by tactile sensors (Lin et al., 2022; Melnik et al., 2021), and the "object is placed" indicator can be obtained by forward kinematics, visual detection/tracking techniques (Kalashnikov et al., 2018; 2021), or even large vision-language models (Du et al., 2023).

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
