# A   Task Descriptions

For all tasks, we use consistent setups for state spaces, action spaces, and demonstrations. The state spaces adhere to a standardized template that includes proprioceptive robot state information, such as joint angles and velocities of the robot arm, and, if applicable, the mobile base. Additionally, task-specific goal information is included within the state. Please refer to the ManiSkill paper (Gu et al., 2023) for more details. Below, we present the key details pertaining to the tasks used in this paper.

## A.1   Pick-and-Place

- Stage Indicators:
  - Object is grasped: Both of the robot fingers contact the object, and the impulse (force) at the contact points is non-zero.
  - Object is placed: The distance between the object and the goal position is less than 2.5 cm. This is given by the success signal of the original task, not designed by us.
  - Robot and object are stationary: The joint velocities of all robot joints are less than 0.2 rad/s. The object velocity is less than 3 cm/s. *This is given by the success signal of the original task, not designed by us.*
- Object Set: The objects in training tasks are from the YCB dataset, including 74 objects. And the objects in test tasks are from the EGAD dataset, including around 1600 objects.
- Action Space: Delta position of the end-effector and the joint positions of the gripper.
- Demonstrations: We use 100 demonstration trajectories in total for this task family (around 1.4 trajectories per task). The demonstrations are from a trained RL agent.

## A.2   Turn Faucet

- Stage Indicators:
  - Handle is moving: The joint velocity of the target joint is greater than 0.01 rad/s.
  - Handle reached the target angle: The joint angle is greater than 90% of the limit. *This is given by the success signal of the original task, not designed by us.*
- Object Set: The objects in training and test tasks are both from the PartNet-Mobility dataset. The training tasks include 10 faucets, and the test tasks include 50 faucets.
- Action Space: Delta pose of the end-effector and joint positions of the gripper.
- Demonstrations: We use 100 demonstration trajectories in total for this task family (around 10 trajectories per task). The demonstrations are from a trained RL agent.

## A.3   Open Cabinet Door

- Stage Indicators:
  - Handle is grasped: Both of the robot fingers contact the handle, and the impulse (force) at the contact points is non-zero.
  - Door is open enough: The joint angle is greater than 90% of the limit. This is given by the success signal of the original task, not designed by us.
  - Door is stationary: The velocity of the door is less than 0.1 m/s, and the angular velocity is less than 1 rad/s. *This is given by the success signal of the original task, not designed by us.*
- Object Set: The objects in training and test tasks are both from the PartNet-Mobility dataset. The training tasks include 4 cabinet doors, and the test tasks include 6 cabinet doors. We remove all single-door cabinets in this task family, as they can be solved by kicking the side of the door and this behavior can be readily learned by sparse rewards.
- Action Space: Joint velocities of the robot arm joints and mobile robot base, and joint positions of the gripper.
- Demonstrations: We use 200 demonstration trajectories in total for this task family (around 50 trajectories per task). The demonstrations are from a trained RL agent.

## B COMPARISON OF HUMAN EFFORT: STAGE INDICATORS VS. HUMAN-ENGINEERED REWARDS

This section explains why designing stage indicators is much easier than designing a full dense reward.

The key challenges in reward engineering lies in **designing reward candidate terms** and **tuning associated hyperparameters**. To illustrate, let us use the "Open Cabinet Door" task familly as an example. The code of human engineered reward is in Listing 1, and the code of our stage indicators is in Listing 2.

The human-engineered reward involves the following reward candidate terms:

- Distance between the robot gripper quaternion, and a set of manually designed grasp quaternions
- Distance between robot hand and door handle
- Signed-distance between tool center point (center of two fingertips) and door handle
- Robot joint velocity
- Door handle velocity
- Door handle angular velocity
- Door joint velocity
- Door joint position
- Multiple boolean functions to determine task stages

Each reward candidate term needs 1∼4 hyperparameters (e.g., normalization function, clip upper bound, clip lower bound, scaling coefficient). In total, this reward function involves **more than 20 hyperparameters** to tune. *The major effort of reward engineering is thus spent iterating over these candidate terms and tuning the hyperparameters by trail and error.* This process is laborious but critical for the success of human-engineered rewards. According to the authors of ManiSkill, they spend **over one month** crafting the dense reward for the "Open Cabinet Door" tasks.

In contrast, our stage indicators for "Open Cabinet Door" tasks only requires to design **two boolean functions**: whether the robot has grasped the handle and whether the door is open enough. The third stage indicator is given by the tasks success signal so we do not need to design it. This trims the number of hyperparameters down from 20+ to just 1 (the first boolean function requires one hyperparameter, and the second boolean function is directly taken from the task's success condition so no hyperparamters), and reduces the lines of code from 100+ to 7 (with a utility function to check grasping, which is from the original codebase).

Therefore, our approach significantly reduces the human effort required for reward engineering.

```
1    def _compute_grasp_poses(self, mesh: trimesh.Trimesh, pose: sapien.
     Pose):
2        # NOTE(jigu): only for axis-aligned horizontal and vertical cases
3        mesh2: trimesh.Trimesh = mesh.copy()
4        # Assume the cabinet is axis-aligned canonically
5        mesh2.apply_transform(pose.to_transformation_matrix())
6
7        extents = mesh2.extents
8        if extents[1] > extents[2]:  # horizontal handle
9            closing = np.array([0, 0, 1])
10       else:  # vertical handle
11           closing = np.array([0, 1, 0])
12
13       # Only rotation of grasp poses are used. Thus, center is dummy.
14       approaching = [1, 0, 0]
15       grasp_poses = [
16           self.agent.build_grasp_pose(approaching, closing, [0, 0, 0]),
17           self.agent.build_grasp_pose(approaching, -closing, [0, 0, 0])
     ,
18       ]
```

```python
19
20          pose_inv = pose.inv()
21          grasp_poses = [pose_inv * x for x in grasp_poses]
22
23          return grasp_poses
24
25      def _compute_handles_grasp_poses(self):
26          self.target_handles_grasp_poses = []
27          for i in range(len(self.target_handles)):
28              link = self.target_links[i]
29              mesh = self.target_handles_mesh[i]
30              grasp_poses = self._compute_grasp_poses(mesh, link.pose)
31              self.target_handles_grasp_poses.append(grasp_poses)
32
33      def compute_dense_reward(self, *args, info: dict, **kwargs):
34          reward = 0.0
35
36          # ----------------------------------------------------- #
37          # The end-effector should be close to the target pose
38          # ----------------------------------------------------- #
39          handle_pose = self.target_link.pose
40          ee_pose = self.agent.hand.pose
41
42          # Position
43          ee_coords = self.agent.get_ee_coords_sample()  # [2, 10, 3]
44          handle_pcd = transform_points(
45              handle_pose.to_transformation_matrix(), self.
    target_handle_pcd
46          )
47          # trimesh.PointCloud(handle_pcd).show()
48          disp_ee_to_handle = sdist.cdist(ee_coords.reshape(-1, 3),
    handle_pcd)
49          dist_ee_to_handle = disp_ee_to_handle.reshape(2, -1).min(-1)   #
    [2]
50          reward_ee_to_handle = -dist_ee_to_handle.mean() * 2
51          reward += reward_ee_to_handle
52
53          # Encourage grasping the handle
54          ee_center_at_world = ee_coords.mean(0)  # [10, 3]
55          ee_center_at_handle = transform_points(
56              handle_pose.inv().to_transformation_matrix(),
    ee_center_at_world
57          )
58          # self.ee_center_at_handle = ee_center_at_handle
59          dist_ee_center_to_handle = self.target_handle_sdf.signed_distance
    (
60              ee_center_at_handle
61          )
62          # print("SDF", dist_ee_center_to_handle)
63          dist_ee_center_to_handle = dist_ee_center_to_handle.max()
64          reward_ee_center_to_handle = (
65              clip_and_normalize(dist_ee_center_to_handle, -0.01, 4e-3) - 1
66          )
67          reward += reward_ee_center_to_handle
68
69          # pointer = trimesh.creation.icosphere(radius=0.02, color=(1, 0,
    0))
70          # trimesh.Scene([self.target_handle_mesh, trimesh.PointCloud(
    ee_center_at_handle)]).show()
71
72          # Rotation
73          target_grasp_poses = self.target_handles_grasp_poses[self.
    target_link_idx]
74          target_grasp_poses = [handle_pose * x for x in target_grasp_poses
    ]
```

```
75          angles_ee_to_grasp_poses = [
76              angle_distance(ee_pose, x) for x in target_grasp_poses
77          ]
78          ee_rot_reward = -min(angles_ee_to_grasp_poses) / np.pi * 3
79          reward += ee_rot_reward
80
81          # ------------------------------------------------- #
82          # Stage reward
83          # ------------------------------------------------- #
84          coeff_qvel = 1.5  # joint velocity
85          coeff_qpos = 0.5  # joint position distance
86          stage_reward = -5 - (coeff_qvel + coeff_qpos)
87          # Legacy version also abstract coeff_qvel + coeff_qpos.
88
89          link_qpos = info["link_qpos"]
90          link_qvel = self.link_qvel
91          link_vel_norm = info["link_vel_norm"]
92          link_ang_vel_norm = info["link_ang_vel_norm"]
93
94          ee_close_to_handle = (
95              dist_ee_to_handle.max() <= 0.01 and dist_ee_center_to_handle
    > 0
96          )
97          if ee_close_to_handle:
98              stage_reward += 0.5
99
100             # Distance between current and target joint positions
101             # TODO(jigu): the lower bound 0 is problematic? should we use
     lower bound of joint limits?
102             reward_qpos = (
103                 clip_and_normalize(link_qpos, 0, self.target_qpos) *
    coeff_qpos
104             )
105             reward += reward_qpos
106
107             if not info["open_enough"]:
108                 # Encourage positive joint velocity to increase joint
    position
109                 reward_qvel = clip_and_normalize(link_qvel, -0.1, 0.5) *
    coeff_qvel
110                 reward += reward_qvel
111             else:
112                 # Add coeff_qvel for smooth transition of stagess
113                 stage_reward += 2 + coeff_qvel
114                 reward_static = -(link_vel_norm + link_ang_vel_norm *
    0.5)
115                 reward += reward_static
116
117                 # Legacy version uses static from info, which is
    incompatible with MPC.
118                 # if info["cabinet_static"]:
119                 if link_vel_norm <= 0.1 and link_ang_vel_norm <= 1:
120                     stage_reward += 1
121
122         # Update info
123         info.update(ee_close_to_handle=ee_close_to_handle, stage_reward=
    stage_reward)
124
125         reward += stage_reward
126         return reward
```

Listing 1: Human-engineered rewards for Open Cabinet Door tasks. The code is from the ManiSkill2 github repo (commit id: 493be36).

```
1 def compute_stage_indicators(self):
```

```
2    stage_indicators = [
3        self.agent.check_grasp(self.target_link), # this utility function
     is given by the original ManiSkill2 codebase. It requires one
    hyperparameter 'max_angle' but we just use the default value
4        self.link_qpos >= self.target_qpos, # door is open enough
5        # the 3rd stage indicator is just the task success signal, so we
    don't need to include it here
6    ]
7    for i in range(1, len(stage_indicators)):
8        stage_indicators[i-1] |= stage_indicators[i]
9    return stage_indicators
```

Listing 2: Our stage indiactors for Open Cabinet Door tasks which is way more easier to design than the human-engineered rewards.

## C    COMPARISON WITH TEXT2REWARD

Text2Reward (Xie et al., 2023) is a concurrent work with our paper. We offer a comparison in this section to help readers understand the differences between our paper and Text2Reward.

While both (Xie et al., 2023) and our paper share the common goal of generating rewards for new tasks, they employ fundamentally distinct setups and methodologies. In short, the primary distinction lies in the fact that **our approach learns rewards from training tasks and success signals (or stage indicators), while (Xie et al., 2023) generates rewards based on exemplar reward codes and the knowledge embedded in Large Language Models (LLMs)**.

To elaborate, the following disparities exist in respective setups and assumptions:

- Both (Xie et al., 2023) and our methods need to interact with environments. However, we emphasize more on evaluating the learned rewards on unseen test tasks.
- (Xie et al., 2023) assumes access to a pool of instruction-reward code pairs, while our method requires training on relevant training tasks instead.
- (Xie et al., 2023) assumes access to the source code of the tasks, allowing them to provide LLMs with a Pythonic environment abstraction and various utility functions. In contrast, our method solely relies on success signals (or stage indicators) and does not require the code of the tasks.

## D    IMPLEMENTATION DETAILS

### D.1    REWARD LEARNING PHASE

#### D.1.1    NETWORK ARCHITECTURES

- Actor Network: 4-layer MLP, hidden units (256, 256, 256)
- Critic Networks: 4-layer MLP, hidden units (256, 256, 256)
- Discriminator Networks (Reward): 2-layer MLP, hidden units (32)

#### D.1.2    HYPERPARAMETERS

We use SAC (Haarnoja et al., 2018) as the backbone RL algorithm in the reward learning phase of DrS. The related hyperparameters are listed in Table 1.

### D.2    REWARD REUSE PHASE

#### D.2.1    NETWORK ARCHITECTURES

- Actor Network: 4-layer MLP, hidden units (256, 256, 256)
- Critic Networks: 4-layer MLP, hidden units (256, 256, 256)

### D.2.2 HYPERPARAMETERS

During the reward reuse phase, we use different rewards to train agents by SAC (Haarnoja et al., 2018). The related hyperparameters are listed in Table 2.

| Name | Value |
| --- | --- |
| replay buffer size | $+\infty$ |
| update-to-data (UTD) ratio | 0.5 |
| optimizer | Adam |
| actor learning rate | 3e-4 |
| critic learning rate | 3e-4 |
| discriminator learning rate | 3e-4 |
| target smoothing coefficient | 0.005 |
| discount factor | 0.8 |
| training frequency | 64 steps |
| target network update frequency | 1 step |
| discriminator update frequency | 1 step |
| batch size | 1024 |
| Auto-tune Entropy | True |

Table 1: The hyperparameters used in reward learning phase of DrS.

| Name | Value |
| --- | --- |
| replay buffer size | $+\infty$ |
| update-to-data (UTD) ratio | 0.5 |
| optimizer | Adam |
| actor learning rate | 3e-4 |
| critic learning rate | 3e-4 |
| target smoothing coefficient | 0.005 |
| discount factor | 0.8 |
| training frequency | 64 steps |
| target network update frequency | 1 step |
| batch size | 1024 |
| Auto-tune Entropy | True |

Table 2: The hyperparameters used in the reward reuse phase of DrS.

## E ADDITIONAL ABLATION STUDY

In this section, we present more ablation studies that are not included in the main paper due to the space limit. These experiments are conducted on the Pick-and-Place task family.

### E.1 MODALITY OF THE INPUTS TO THE REWARDS

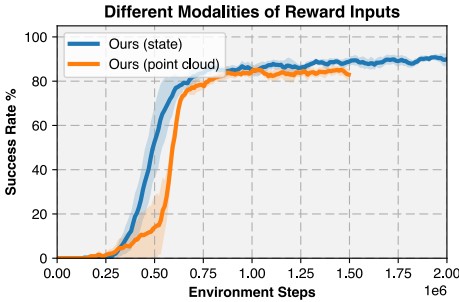

Figure 9: An experiment about using our approach with point cloud inputs, and the point clouds are processed by a PointNet.

Our approach is able to accommodate various input modalities for the reward functions, including both low-dimensional state vectors and high-dimensional visual inputs. To demonstrate this compatibility, we conducted an additional experiment using point cloud inputs. In this experiment, the reward function (discriminator) not only considers the low-dimensional state but also takes a point cloud as input, with the point cloud being processed by a PointNet. The results of this experiment are depicted in Fig. 9.

We can see that the reward with point cloud input performs similarly to the one with state input, which shows that our approach is perfectly compatible with high-dimensional visual inputs. However, the techniques about visual inputs are a bit orthogonal to our focus (reward learning), and learning with visual inputs takes significantly more time, so we still keep most of our experiments on state inputs.

The results reveal that the reward function utilizing point cloud input performs comparably to the one utilizing state input, demonstrating the seamless integration of our approach with high-dimensional visual inputs. However, it is worth noting that the techniques about visual inputs, while compatible with our framework, are a little bit orthogonal to our focus (reward learning). Moreover, learning with visual inputs typically takes a significantly longer training time. Consequently, the majority of our experiments primarily use state inputs, allowing us to concentrate on the core aspects of reward learning.

## E.2 DISCRIMINATOR MODIFICATION AND STAGE INDICATORS

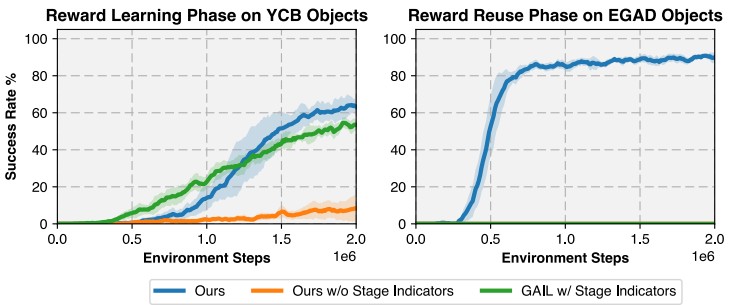

Figure 10: An ablation study was conducted to examine the impact of discriminator modification and stage indicators. Both the reward learning phase and reward reuse phase are shown. The learned rewards from the ablated baselines failed to successfully train new agents in the test tasks.

In contrast to GAIL (Ho & Ermon, 2016), our approach incorporates two critical modifications in the training of discriminators to facilitate the learning of reusable dense rewards. These modifications entail: (a) replacing the agent-demonstration discriminator with the success-failure discriminator, and (b) employing stage indicators by utilizing a separate discriminator for each stage. To ascertain the significance of these modifications, we devised two ablation baselines:

- **GAIL w/ Stage Indicators**: This baseline serves as an equivalent representation of our method without the incorporation of the success-failure discriminator. In GAIL, the discriminator solely distinguishes between agent and expert trajectories, making it incapable of learning separate rewards for each stage. To incorporate the stage indicators within the GAIL framework, we first train the original GAIL on the training tasks. During the reward reuse phase, we linearly combine the GAIL reward with the semi-sparse reward, thus leveraging the stage information. Through experimentation, we explored different weightings to strike an optimal balance between these two reward components.

- **Ours w/o Stage Indicators**: In this baseline, we exclude the stage indicators and solely rely on the task completion signal to train the discriminator. This approach is equivalent to the one-stage reward learning discussed in Sec. 4.1.

Fig. 10 illustrates the comparison between the two ablation baselines and our method during both the reward learning phase and reward reuse phase. While both "GAIL w/ Stage Indicators" and "Ours w/o Stage Indicators" demonstrate similar success rates as our method at the conclusion of the reward learning phase, it is crucial to emphasize that the learned rewards from both ablation baselines **fail** to be reused to the test tasks. In contrast, our method achieves the acquisition of high-quality reward functions capable of effectively training new RL agents in the test tasks. This outcome substantiates the indispensability of the two proposed components in facilitating the acquisition of reusable dense rewards.

## E.3 REWARD FORMULATION

In our approach, we leverage the stage indicators and define the reward function as the sum of the semi-sparse reward and the discriminator's bounded prediction for the current stage, as expressed in Eq. 4. This formulation ensures that the reward strictly increases across stages. To evaluate

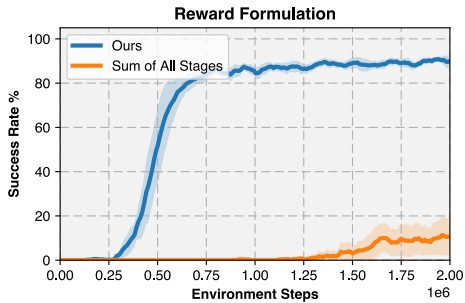

Figure 11: Ablation study of reward formulation. Comparison is done by reusing the learned rewards to train new agents on the test tasks.

the effectiveness of this formulation, we compare it with a straightforward variant, denoted as $\sum_k \tanh(\text{Discriminator}_k(s'))$, which sums up the discriminator predictions for all stages. As depicted in Fig. 11, the simple variant exhibits significantly poorer performance, underscoring the importance of focusing on the dense reward specific to the current stage.

## F   AUTOMATICALLY GENERATING STAGE INDICATORS

This section discusses a few promising solutions to automatically generate stage indicators, drawing inspiration from some recent publications. Though this topic is *a little bit beyond the scope of our paper*, we believe this is a valuable discussion for the readers.

### F.1   EMPLOY LLMs FOR CODE GENERATION OF STAGE INDICATORS

Beyond task decomposition, LLMs demonstrate the capability to directly write code (Liang et al., 2023; Singh et al., 2023; Yu et al., 2023; Ha et al., 2023) for robotic tasks. A recent study (Ha et al., 2023) exemplified how LLMs, when prompted with the appropriate APIs, can generate success conditions (code snippets) for each subtask. Given the swift advancements in the domain of large models, it is entirely feasible to generate both stage structures and stage indicators using them.

### F.2   INFER STAGES VIA KEYFRAME DISCOVERY

The boundaries between stages can be viewed as keyframes in the trajectories. A recent approach introduced by (Shi et al., 2023) suggests the automated extraction of such keyframes from trajectories, leveraging reconstruction errors. Given these keyframes, one intuitive solution is to develop a keyframe classifier that can act as a stage indicator. However, this requires a certain degree of consistency across keyframes, and we believe it is an interesting direction to explore.

# G    ADDITIONAL EXPERIMENTS ON OTHER DOMAINS

## G.1    NAVIGATION

### G.1.1    INTRODUCTION

In this section, we incorporated experiments on navigation tasks, which were conducted during the initial stages of our project. We do not include these results in the main paper, as we found these simple navigation tasks to be less interesting compared to the robot manipulation tasks.

### G.1.2    SETUP

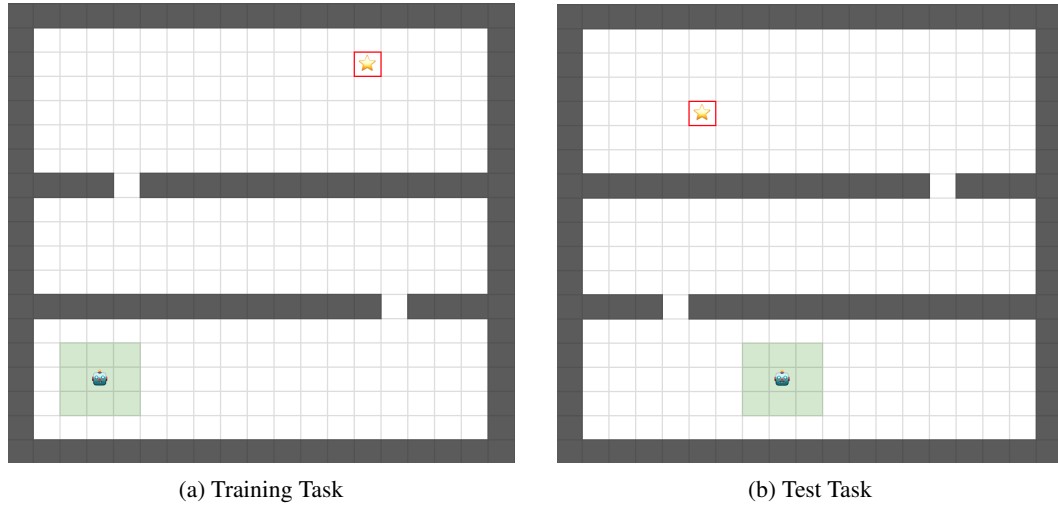

(a) Training Task                              (b) Test Task

Figure 12: Visualization of the training and test tasks in the navigation domain. The agent begins at a random location in the bottom room, and the goal is randomly positioned in the top room.

**Task Description**    We have developed a 2D navigation task conceptually similar to MiniGrid, as visually represented in Fig. 12. The maps are 17x17, where the agent is randomly placed in the bottom room and needs to navigate to the star, randomly located in the top room.

**Observation**    Observations provided to the agent include its xy coordinates, the xy coordinates of the goal, and a 3x3 patch around itself.

**Action**    The agent has a choice of 5 actions: moving up, down, left, right, or remaining stationary.

**Training and Test Set**    The reward is learned on the map shown in 12a and then reused on the map in 12b. The difference between these two maps lies in the positions of two gates.

### G.1.3    RESULTS

Our method is also effective in learning reusable rewards for navigation tasks. Given the relative simplicity of this specific navigation task, our approach's one-stage version suffices, eliminating the need for additional stage information. The results for this experiment are shown in Fig. 13.

The results clearly demonstrate that the learned reward from our approach successfully guides the RL agent to complete the task perfectly. In contrast, RL agents with sparse rewards show poor performance. Note that the map used in the test task differs from the training one, so directly transferring policy would not work. We also visualize the learned reward in Fig. 14. See the caption for a detailed analysis.

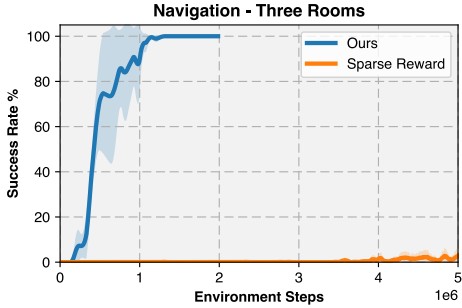

Figure 13: Evaluation results of reusing learned rewards in the navigation task. All curves use DQN to train, but with different rewards. 3 random seeds, the shaded region is std.

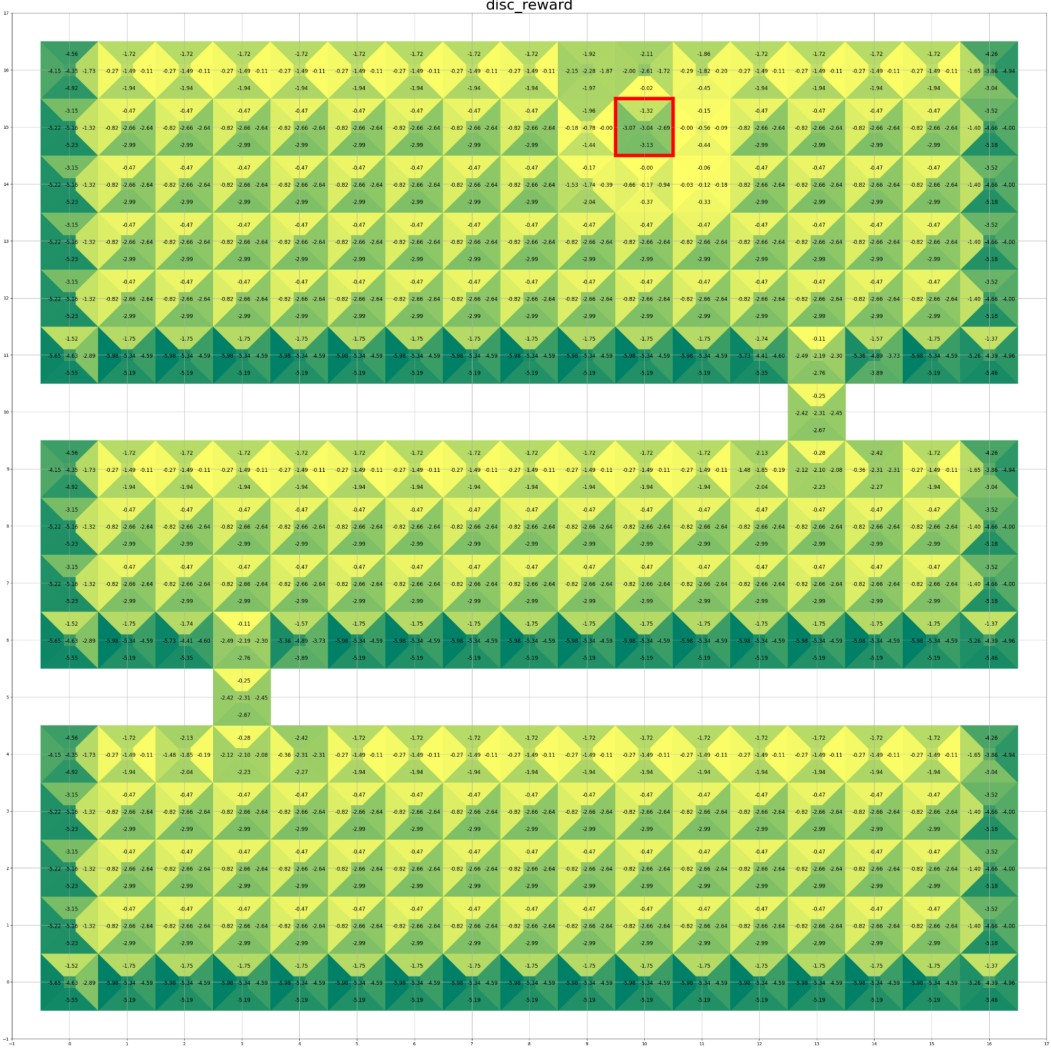

Figure 14: Visualization of Learned Reward: Each cell displays five values corresponding to the rewards for five different actions. A lighter color means a higher reward value. The red box shows the location of a randomly chosen goal. Note that the inputs to both the learned reward function and the agent include only the local 3x3 area around the agent, excluding any information about the gate positions. Overall, the learned reward encourages upward movement, aligning with the placement of the goal in the top room. When the gates are not within the agent's local 3x3 patch, the rewards for moving left or right are nearly equivalent, which is reasonable since the gate's position cannot be determined without direct observation. However, when the gates are visible within the local patch, the learned reward directs the agent to go through these gates. This behavior of the learned reward aligns well with the task's objectives.

## G.2 LOCOMOTION

### G.2.1 INTRODUCTION

While it can be tricky to divide locomotion tasks into stages, our method (specifically, the one-stage version) is capable of effectively handling such tasks, if they have a short horizon. In this section, we demonstrate that our approach can learn reusable rewards for Half Cheetah, a representative locomotion task in MuJoCo.

For tasks that are long-horizon and hard to specify stages, such as the Ant Maze, crafting rewards is very challenging even for experienced human experts. Therefore, we leave these tasks for future work.

### G.2.2 SETUP

**Task Description** Our experiment uses `HalfCheetah-v3` from Gymnasium. The `HalfCheetah` task has a predefined reward threshold of 4800, as specified in their code, which is used to gauge task completion according to their documentation. Thus, we define the sparse reward (success signal) for this task as achieving an accumulative dense reward greater than 4800.

**Training and Test Set** In the reward learning phase, we use the standard `HalfCheetah-v3` task. In the reward reuse phase, we modify the task by increasing the damping of the front leg joints (thigh, shin, and foot joints) by 1.5 times. This increased damping makes it more challenging for the cheetah to achieve high speeds.

### G.2.3 RESULTS

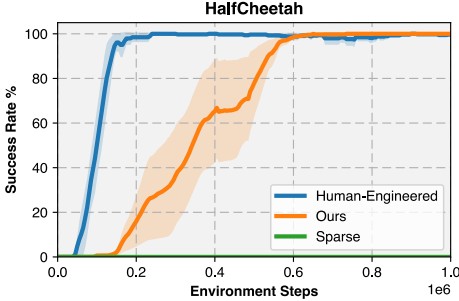

Figure 15: Evaluation results of reusing learned rewards in the `HalfCheetah-v3` task. All curves use SAC to train, but with different rewards. 3 random seeds, the shaded region is std.

Our method has successfully demonstrated its capability to learn reusable rewards in the Half Cheetah task. The results are illustrated in Fig. 15. Notably, the performance achieved using the learned reward is comparable to that of the human-engineered reward, while the sparse reward proved ineffective in training an RL agent. Given that many locomotion tasks emphasize low-level control and are typically of a shorter horizon, our approach's one-stage version proves to be highly effective. Additionally, this version does not require any stage information, further underscoring its efficiency and adaptability in handling such tasks.

# H    DISCUSSION ON THE DESIRED PROPERTIES OF DENSE REWARDS

## H.1    OVERVIEW

Our paper primarily focuses on learning a dense reward, so one important question we want to discuss is: *What kind of dense reward do we aspire to learn?*

It is somewhat challenging to strictly distinguish dense rewards from sparse rewards, due to the lack of strict definitions of dense rewards in the existing literature (to the best of our knowledge). However, this does not preclude a meaningful discussion about the desired properties of dense rewards. Unlike sparse rewards, which typically only provide reward signals when the task is solved, dense rewards offer more frequent and immediate feedback regarding the agent's actions.

We posit that the fundamental property of an *effective* dense reward is its capacity to enhance the sample efficiency of RL algorithms. The rationale behind this property is straightforward: a well-structured dense reward should **reduce the need for extensive exploration during RL training**. By providing direct guidance and immediate feedback, the agent can quickly discover optimal actions, thereby accelerating the learning process.

In line with this philosophy, an **ideal dense reward should allow the derivation of optimal policies with minimal effort**. By analyzing a simple tabular case, we find that **our learned reward exhibits this great property**. To be more specific, in the example below, **we can obtain the optimal policy by greedily following the path of maximum reward at each step**.

## H.2    ANALYSIS ON A SIMPLE TABULAR CASE

Under certain assumptions, we can obtain the optimal policy by greedily following the path of maximum reward at each step, i.e.,

$$\pi^*(s) = \arg\max_a R^\dagger(s, a)$$

, where $\pi^*$ is the optimal policy and $R^\dagger$ is the learned reward.

### H.2.1    SETUP AND ASSUMPTIONS

In this analysis, we consider a MDP with the following assumptions:

- Deterministic transitions: $s' = P(s, a)$

- Discrete and finite state/action space: $S = \{s_0, s_1, ...\}$, $A = \{a_0, a_1, ...\}$

- Given sparse reward: $R(s, a, s') = 1$ if $s' = s_{goal}$, otherwise 0

- Discount factor: $\gamma < 1$

Other assumptions about our approach:

- Only one stage, so the one-stage version of our approach is applied.

- The buffers for success trajectories and failure trajectories are large enough, but not infinite.

- After training for a sufficiently long time, policy converges to the optimal policy $\pi^*$. (This is a strong assumption, but it is possible in theory.)

### H.2.2    NOTATIONS

- Learned reward: $R^\dagger(s, a) = \tanh(\text{Discriminator}(s, a))$, so $R^\dagger(s, a) \in (-1, 1)$

- Buffer for success trajectories $\mathcal{B}^+$, buffer for failure trajectories $\mathcal{B}^-$

- Optimal policy: $\pi^*(a|s)$, which represents the probability of choosing action $a$ at state $s$. Here we overload the notation $\pi^*$ to capture the potential multi-modal output of the policy.

### H.2.3  CONNECTION BETWEEN OPTIMAL POLICY AND LEARNED REWARD

Here, we want to demonstrate that *the learned reward of an optimal action is always higher than that of any non-optimal action in each state*. If this holds, it then becomes feasible to straightforwardly identify the optimal action at each state by adopting a greedy strategy that selects the action yielding the highest reward.

When $\gamma < 1$, $\pi^*$ will go to $s_{goal}$ by the shortest paths, so $\pi^*(a|s) = 1/k_s$ or $0$, where $k_s$ is the number of optimal actions at $s$.

$\forall s$, there are two kinds of actions $a^+$ and $a^-$

1. $\pi^*(a^+|s) > 0$, which means $a^+$ is one of the optimal actions. Then $(s, a^+)$ must be in $\mathcal{B}^+$, possibly be in $\mathcal{B}^-$. Therefore, $R^\dagger(s, a^+) > -1 + \epsilon$, when the discriminator converges. This is because the buffers are finite-size, $(s, a^+)$ will be sampled into positive training data of the discriminator with a probability larger than 0.

2. $\pi^*(a^-|s) = 0$, which means $a^-$ is NOT one of the optimal actions. Then $(s, a^-)$ will only be in $\mathcal{B}^-$, and will NOT be in $\mathcal{B}^+$ Therefore, $R^\dagger(s, a^-) \to -1$, when the discriminator converges. This is because $(s, a^-)$ will only show in the negative training data of the discriminator.

Therefore, we have $R^\dagger(s, a^+) > R^\dagger(s, a^-)$ for all states $s$. By employing a greedy strategy that selects $\arg\max_a R^\dagger(s, a)$, we can reach the goal states in the same way as how the optimal policy $\pi^*$ reaches the goal.

### H.3  FURTHER DISCUSSIONS

This subsection is dedicated to addressing additional questions the readers may raise after reading the above analysis.

### H.3.1  DOES THE ABOVE CONCLUSION GENERALIZE TO MORE COMPLICATED CASES?

Although our analysis highlights a desirable property of the learned reward in a simple tabular case, this finding should not be hastily generalized to more complex cases, such as the robotic manipulation tasks used in our paper. This caution is due to two primary reasons:

1. In environments where the state and action spaces are continuous, the ability of the neural network to interpolate plays a significant role in shaping the final learned reward.

2. Practically, achieving convergence for both the policy and the discriminator can be a very time-consuming process.

### H.3.2  THE NECESSITY OF LEARNED REWARD DESPITE ITS SIMILARITY TO POLICY

The learned reward might appear redundant at first glance, as it seems to convey the same information as the learned policy. This observation raises a potential question: why is there a need for a learned reward if we already have a learned policy? Couldn't we just utilize the learned policy directly?

The answer lies in the distinct advantages that the learned reward offers, particularly when adapting to new tasks. When the environment dynamics change, a new policy can be effectively retrained using the learned reward in conjunction with the new environmental dynamics. Directly transferring the policy, or fine-tuning it with a sparse reward, can be less efficient in certain situations. For a practical illustration of this concept, refer to Fig. 14 and Sec, G.1.3. These sections provide a compelling example where the transfer of rewards demonstrates success, in contrast to the less effective transfer of policies.