# OpenReview forum: "DrS: Learning Reusable Dense Rewards for Multi-Stage Tasks"
_ICLR.cc/2024/Conference — ICLR 2024 poster_

### Official Review · Reviewer_35ZH · 2023-10-19

**Soundness:** 2 fair
**Presentation:** 3 good
**Contribution:** 2 fair
**Rating:** 6
**Confidence:** 4

**Summary:**

This paper considers learning reusable dense rewards for multi-stage tasks and proposes a method named as DrS. DrS classifies each collected trajectory by its stage and relabels them into failure and success ones. It then defines the learned reward function as the stage index plus the output of the corresponding discriminator. Extensive experiments from three task families showcase the effectiveness of DrS.

**Strengths:**

1. This paper is clearly written and easy to follow.
2. Authors conducted extensive experiments over 1000 task variants from threes task families to empirically evaluate their proposed method.
3. Learning reusable rewards is an interesting and important topic for RL.

**Weaknesses:**

1. The experimental results are all from the ManiSkill benchmark. Considering other domains such as navigation (say, the Minigrid benchmark) would make the results more convincing.
2.  Lack of discussion on some related work, such as [1, 2].
3. The proposed method heavily relies on the stage structures of the task (see Figure 6, 1-stage fails), but there may exist tasks hard to specify stages (e.g., locomotion tasks of mujoco).
4. Classifying the collected trajectories into success and failure ones and learning a corresponding reward function may not incentive the agent to finish the task optimally since no matter how sub-optimal the trajectory is, as long as it reaches the highest stage, it will be regarded as the most successful one.

[1] Yoo, Se-Wook, and Seung-Woo Seo. "Learning Multi-Task Transferable Rewards via Variational Inverse Reinforcement Learning." 2022 International Conference on Robotics and Automation (ICRA). IEEE, 2022.

[2] Luo, Fan-Ming, Xingchen Cao, and Yang Yu. "Transferable Reward Learning by Dynamics-Agnostic Discriminator Ensemble." arXiv preprint arXiv:2206.00238 (2022).

**Questions:**

1. The authors asked "can we learn dense reward functions in a data-driven manner?" But what kind of dense reward do we really need? Say, there are three reward function $r_1$, $r_2$ and $r_3$, where $r_1=1$ if reaching the goal state, otherwise $r_1=0$; $r_2=0$ if reaching the goal state, otherwise $r_2=-1$; $r_3=2$ if reaching the goal state, otherwise $r_3=1$. Are $r_2$ and $r_3$ your considered dense reward functions and better than $r_1$? Why?
2. How is the training time? DrS has to classify each collected trajectory based on the maximal stage index of all the transitions' stage indexes, which seems to be quite time consuming.
3. Why is GAIL unable to learn reusable rewards? I understand that at convergence, the discriminator outputs 1/2 for every expert state-action transition, but correspondingly, other state-action pairs will be assigned lower values. It seems to be a reasonable dense reward function. In Figure 12, the authors compare DrS with "GAIL w/ stage indicators", but what if GAIL?
4. In line 17 of Algorithm 1's pseudocode, we need to calculate the reward for each transition. But there are multiple discriminators, so which one should we select? Based on the stage index of the trajectory? If so, the same next state may get different rewards because of being in different trajectories. Will it cause the training process unstable?

I am willing to raise my scores if you could sovle my concerns.

---

> ### Author Response · Authors · 2023-11-15
> **Author Response (1/3)**
>
> Thank you for carefully reviewing our manuscript and offering insightful feedback! We are delighted to know that you find our **topic important, our experiments extensive, and our paper is clearly written**. In response to your comments, we have addressed each concern as follows.
>
> *Note: The outcomes of additional experiments have been added to Appendix G. We kindly request you to download the latest version of the PDF file. Following the conclusion of the discussion phase, we will further update the manuscript to incorporate key discussions.*
>
> ---
>
> ## Responses to Weaknesses
>
> > The experimental results are all from the ManiSkill benchmark. Considering other domains such as navigation (say, the Minigrid benchmark) would make the results more convincing.
>
> Thank you for your valuable suggestion! Our primary focus has been on manipulation tasks due to their significant real-world applicability and the complexity involved in designing their reward structures. Nonetheless, we acknowledge that extending our experiments to include other domains can provide a more thorough evaluation of our method. Interestingly, our project initially began with some simple navigation tasks before progressing to the more challenging manipulation tasks.
>
> In Appendix G, we present our results in two additional domains: navigation and locomotion. Briefly, our method demonstrates effective performance in both navigation and locomotion tasks, significantly surpassing the baselines that use sparse rewards. We will also discuss the locomotion results in a later part of our response.
>
> ---
>
> > Lack of discussion on some related work, such as [1, 2].
>
> Thank you for suggesting these relevant studies. References [1,2] are both great papers and we **will cite them** in the revised manuscript post the discussion phase.
>
> In comparing our work with [1], we note that their research delves into multi-task learning and multi-task transfer learning via variational inverse RL, with a focus on tasks that involve rearranging a set of subtasks in different orders. In contrast, our study primarily explores the reuse of rewards in novel tasks involving different objects, dynamics, action spaces, and robot morphologies.
>
> Regarding [2], the paper presents a method for learning rewards that reduces the reliance on both dynamics and policy, centering specifically on rewards conditioned on both actions and states. Our approach, while applicable to different reward inputs, has been mainly concentrated on state-based rewards, as these are widely used in many modern robot learning benchmarks [3,4,5,6,7,8].
>
> [3] Zhu, Yuke, et al. "robosuite: A modular simulation framework and benchmark for robot learning." arXiv preprint arXiv:2009.12293 (2020).
>
> [4] Gu, Jiayuan, et al. "ManiSkill2: A Unified Benchmark for Generalizable Manipulation Skills." International Conference on Learning Representations.
>
> [5] Mu, Tongzhou, et al. "ManiSkill: Generalizable Manipulation Skill Benchmark with Large-Scale Demonstrations." Thirty-fifth Conference on Neural Information Processing Systems Datasets and Benchmarks Track.
>
> [6] Rajeswaran, Aravind, et al. "Learning complex dexterous manipulation with deep reinforcement learning and demonstrations." arXiv preprint arXiv:1709.10087 (2017).
>
> [7] Yu, Tianhe, et al. "Meta-world: A benchmark and evaluation for multi-task and meta reinforcement learning." Conference on robot learning. PMLR, 2020.
>
> [8] Mittal, Mayank, et al. "ORBIT: A Unified Simulation Framework for Interactive Robot Learning Environments." arXiv preprint arXiv:2301.04195 (2023).
>
> ---
>
> > The proposed method heavily relies on the stage structures of the task (see Figure 6, 1-stage fails), but there may exist tasks hard to specify stages (e.g., locomotion tasks of mujoco).
>
> That is a really good point! However, our reward learning approach requires the stage information mainly because the tasks are **long-horizon**. For the tasks that are hard to specify stages, let’s discuss them in two cases:
>
>
> 1. Short-Horizon Tasks: Our method (one-stage version) effectively handles short-horizon tasks without requiring stage information. For example, our approach can learn reusable rewards for HalfChettah (a representative locomotion task in MuJoCo). See Appendix G.2 for details.
>
> 2. Long-Horizon Tasks: When a task is long-horizon and its stages are not easily definable, it presents a significant challenge. Even for human experts, designing a dense reward for such tasks is very challenging. Take the [Ant Maze](https://robotics.farama.org/envs/maze/ant_maze/) task as an example, which is long-horizon and hard to specify stages. The [default dense reward designed by humans](https://github.com/Farama-Foundation/Gymnasium-Robotics/blob/main/gymnasium_robotics/envs/maze/ant_maze_v4.py#L154) is merely the negative L2 distance, which is very bad since it ignores the walls in the maze. Therefore, we leave these tasks for future work.

---

> > ### Author Response · Authors · 2023-11-15
> > **Author Response (2/3)**
> >
> > > Classifying the collected trajectories into success and failure ones and learning a corresponding reward function may not incentive the agent to finish the task optimally since no matter how sub-optimal the trajectory is, as long as it reaches the highest stage, it will be regarded as the most successful one.
> >
> > This point depends on how you define “optimality”.
> >
> > In our paper, we focus on MDPs that lack dense rewards. In such scenarios, all trajectories that successfully reach the goal state are deemed equally "optimal", as they all get the sparse reward.
> >
> > From another perspective, there is a (kind of) “universal” optimality metric: the length of the trajectory. Commonly, the quicker the task completion, the better. If we adopt trajectory length as a metric of optimality, it has been implicitly accounted for by using a discount factor less than 1.
> >
> > Please let us know if you have a different definition of “optimality” in mind and we are happy to further discuss it.
> >
> > ---
> >
> > ## Responses to Questions
> >
> > > … Are $r_2$ and $r_3$ your considered dense reward functions and better than $r_1$? Why?
> >
> > Great question! It is a little bit tricky to strictly define the sparse reward and dense reward.  Conventionally, **dense rewards should provide different feedback when the agents make different decisions**. In your case, all three rewards —$r_1$, $r_2$, and $r_3$—offer uniform feedback at all non-goal states, thus they are not regarded as dense rewards in our context. In practice, they should work similarly if we handle the bellman equation bootstrapping appropriately.
> >
> > The key to effective dense rewards, in our view, is their potential to enhance the sample efficiency of RL algorithms. While it would be intriguing to develop a theoretical framework to evaluate the quality of dense rewards based on sample efficiency, such an exploration is somewhat beyond the scope of this paper.
> >
> > ---
> >
> > > How is the training time? DrS has to classify each collected trajectory based on the maximal stage index of all the transitions' stage indexes, which seems to be quite time consuming.
> >
> > **Reward Learning Phase of DrS:** In this phase, our method requires training $k$ additional discriminators ($k=2$ or $3$ in our experiments). While this adds to the training cost compared to standard RL training, the increase in time is relatively minimal. This efficiency is due to the small network size used for discriminators (a 2-layer MLP with a width of 32). To provide a concrete comparison, here's a table showing the training wall-time for the Pick and Place tasks:
> >
> > | Algorithm | Environment Steps | Wall Time |
> > |-----------|-------------------|-----------|
> > | DrS       | 3M                | ~18 hours |
> > | SAC       | 3M                | ~15 hours |
> >
> >
> > **Reward Reuse Phase of DrS:** In this phase, the primary additional step, compared to regular RL training, is obtaining rewards through a simple network forwarding pass. This step is not time-consuming and does not significantly add to the overall training time.
> >
> > ---
> >
> > > Why is GAIL unable to learn reusable rewards? … In Figure 12, the authors compare DrS with "GAIL w/ stage indicators", but what if GAIL?
> >
> > The discriminator in GAIL is trained to distinguish between nearly identical positive and negative data, especially in the latter stages of training. Even though this discrimination task is impossible to complete, the discriminator will try to distinguish them by overfitting to irrelevant features. Such an overfitted discriminator does not necessarily assign lower values on bad transitions, given that these bad transitions have been absent from its training data for a long time. This phenomenon is discussed in [9], and [10] similarly argues that a GAIL discriminator is unsuitable as a reward. Moreover, [11] highlights that an overfitted network's generalization performance can be arbitrarily bad if the training data labels are ambiguous or randomly assigned.
> >
> > In Fig. 12, we have shown the performance of “GAIL w/ stage indicators”, which is a stronger version of GAIL reward since it utilizes the stage indicators. We have also tried the reward from GAIL and *it yields zero success rates either*.
> >
> > [9] Danfei Xu and Misha Denil. Positive-unlabeled reward learning. In Conference on Robot Learning, pp. 205–219. PMLR, 2021.
> >
> > [10] Justin Fu, Avi Singh, Dibya Ghosh, Larry Yang, and Sergey Levine. Variational inverse control with events: A general framework for data-driven reward definition. Advances in neural information processing systems, 31, 2018.
> >
> > [11] Zhang, Chiyuan, et al. "Understanding deep learning (still) requires rethinking generalization." Communications of the ACM 64.3 (2021): 107-115.

---

> > > ### Author Response · Authors · 2023-11-15
> > > **Author Response (3/3)**
> > >
> > > > … But there are multiple discriminators, so which one should we select?...
> > >
> > > As shown in Eq. 4, we need to use Discriminator $k$, where $k$ is the stage index of the next state $s’$, so $k$ only depends on the next state itself. One state only belongs to one stage (see Fig. 1 for an example), and it will not get different rewards because of being in different trajectories.
> > >
> > > ---
> > >
> > > **Hope the above clarifications and responses address your concerns. If not, feel free to let us know, and we are more than happy to answer additional questions. Thanks so much for your time!**

---

> ### Comment · Reviewer_35ZH · 2023-11-17
> **Thanks to Your Response**
>
> Thanks to your clear and detailed response, which has solved many of my issues. However, I still have some concerns.
>
> > Please let us know if you have a different definition of “optimality” in mind and we are happy to further discuss it.
>
> As in your "Problem Setup" section, you formalize the problem into a discounted MDP with $\gamma$ denoting the discount factor. What's more, in the pseudocode, you use SAC. Therfore, I prefer defining "optimality" as "the quicker the task completion, the better.", which also aligns with the requirements of some real-world goal-conditioned tasks, such as some navigation tasks.
>
> Here, I would like to elaborate on my concerns "...may not incentive the agent to finish the task optimally...". Say, there are two trajectories, $\tau_1$ and $\tau_2$, both reaching the goal state. Since both $\tau_1$ and $\tau_2$ are regarded as the most successful one, the returns calculated from the (finally learned) discriminator will be the same or the longest one gets the highest return since you set $\gamma < 1$. Thus, the learned policy may not try to finish the task as soon as possible.
>
> Altough you could use "success rate" to be the evaluation metric, I still feel that it a limitation.
>
> >....While it would be intriguing to develop a theoretical framework to evaluate the quality of dense rewards based on sample efficiency, such an exploration is somewhat beyond the scope of this paper.
>
> I thank the authors for presenting their answer on my question "… Are $r_2$ and $r_3$ your considered dense reward functions and better than $r_1$? Why?". To some extent, I agree with your answer, "Conventionally, dense rewards should provide different feedback when the agents make different decisions". However, I highly recommend the authors to give more discussion on this question in the paper, which I think is a fundamental question if the authors want to learn "dense" rewards. I do not quite like the phrase "beyond the scope", which sounds like an excuse and does not make any sense.
>
> > it will not get different rewards because of being in different trajectories
>
> I mean, there may be some state occurring in different trajectories. In such case, the same state may get different rewards due to being in different trajectories with different stage index.
>
> Thanks in advance!

---

> ### Author Response · Authors · 2023-11-19
> **2nd Round of Author Response (1/2)**
>
> Thank you for taking the time to read our rebuttal and responding in time! We answer your follow-up questions below.
>
> > … the learned policy may not try to finish the task as soon as possible.
>
> Based on our prior conversation, let's delve deeper into this specific type of "optimality" - "the quicker the task completion, the better." In our setup, the learned policy actually will try to reach the goal states as soon as possible. We offer a short explanation and a longer explanation below.
>
>
> ### **The Short Explanation**
>
> We model all tasks as *continuing tasks*, and the goal states always receive the highest reward. Thus, the optimal policy will learn to reach the goal states as soon as possible, and then remain in these goal states.
>
>
> ### **The Long Explanation**
>
> In our paper, we model all tasks as *continuing tasks* (see Sec. 3.3 in [1] for definition), as opposed to "episodic tasks", when training RL agents. Specifically, we assume that the tasks would not terminate upon reaching the goal. This assumption can be easily achieved in either the environment code or the RL training code (let us know if you would like to know more details about this). Modeling tasks as continuing tasks is a common practice when training RL agents for tasks with positive dense rewards (e.g., MuJoCo control tasks).
>
> According to Eq. 4 in our paper, the reward for goal states is higher than that of all non-goal states. For a visual illustration, refer to Fig. 3 in the paper.
>
> Combining these points, we will see that the optimal policy should reach the goal as soon as possible to get the highest reward, and then stay there.
>
> To clarify this conclusion, consider the MDP shown [here](https://sites.google.com/view/iclr24drs/rebuttal). For simplicity, we assume there is only one stage. States $s_0$, $s_1$, and $s_2$ are in stage 0, while $s_{goal}$ is in stage 1 (stage 1 is not really used and it is not associated with a discriminator).
>
> Based on Eq. 4 in our paper, we can calculate the final learned reward. (assuming $\alpha=\frac{1}{3}$)
> - Reward of $s_1$: $R_1\to\frac{1}{3}$, because $s_1$ always leads to the goal state.
> - Reward of $s_2$: $R_2\to - \frac{1}{3}$, because $s_2$ never leads to the goal state.
> - Reward of $s_{goal}$: $R_{goal}=1$, because the goal state is considered as “stage 1”.
>
> Now, let’s compare the following two successful trajectories:
> - $\tau_1= ( s_0, s_{goal}, s_{goal}, … )$
> - $\tau_2= ( s_0, s_1, s_{goal}, s_{goal}, … )$
>
> Their respective discounted returns are:
> - $G_1=R_{goal} + \gamma^2 R_{goal}+...=\frac{1}{1-\gamma}$
> - $G_2=R_1+\gamma R_{goal} + \gamma^2 R_{goal}+...=\frac{1}{3}+\gamma\frac{1}{1-\gamma}$
>
> Comparing $G_1$ and $G_2$, we calculate the difference: $G_1-G_2=(1-\gamma)\frac{1}{1-\gamma}-\frac{1}{3}=\frac{2}{3}>0$
>
> Thus, $G_1 > G_2$, showing that $\tau_1$ (which reaches the goal faster) yields a higher discounted return.
>
> In practice, the MDP will be more complicated and the learned reward may not fully converge, but we hope the above example illustrates why reaching the goal as soon as possible is preferred by the learned policy in our setup.
>
>
>
> [1] Sutton, Richard S., and Andrew G. Barto. Reinforcement learning: An introduction. MIT press, 2018.
>
> ---
>
> > …I highly recommend the authors to give more discussion on this question in the paper, which I think is a fundamental question if the authors want to learn "dense" rewards.
>
> Thank you for the suggestion! We have included a detailed discussion on the desired properties of dense rewards in **Appendix H** of the updated PDF. We will merge some key discussions into the main paper following the conclusion of the rebuttal phase.
>
> To clarify why we say “beyond the scope” in our first response: Our intention was to convey that
> “developing a theoretical framework to evaluate the quality of dense rewards based on sample efficiency” is beyond the scope of this paper.  To the best of our knowledge, there is very limited literature on the theoretical analysis of dense rewards from a sample efficiency perspective. We believe that formulating such a theoretical framework could constitute a significant contribution to another theoretical paper. Hence, we previously stated that it is “beyond the scope of this paper”. Nevertheless, we do agree that discussing the desired properties of dense rewards is beneficial and have therefore included some discussions in Appendix H for a more rounded understanding.

---

> ### Author Response · Authors · 2023-11-19
> **2nd Round of Author Response (2/2)**
>
> > I mean, there may be some state occurring in different trajectories. In such case, the same state may get different rewards due to being in different trajectories with different stage index.
>
>
> A state can be assigned to different buffers for training the discriminators. However, when computing the learned reward of a state, it will not get different rewards because the *stage index of a state never changes* (the stage index is given by the environment). Eq. 4 in our paper shows how we compute the learned reward for a certain state. Please let us know if you need further clarification on this, we are more than willing to provide additional explanation.
>
> ---
>
> If you feel that our rebuttal has addressed your concerns, we would be grateful if you would consider **revising your score in response**. Thanks so much for your time and effort!

---

> > ### Author Response · Authors · 2023-11-22
> > **Have our responses addressed your concerns?**
> >
> > Dear Reviewer 35ZH,
> >
> > Thank you again for your valuable comments. We hope that our response has addressed your questions and concerns. **The discussion period is coming to an end**, and we would highly appreciate you could let us know if you have any remaining concerns.
> >
> > If you feel that our rebuttal has addressed your concerns, we would be grateful if you would consider **revising your score in response**.
> >
> > Thank you once again for your time, insights, and suggestions!
> >
> > Sincerely,
> >
> > Authors

---

> > > ### Comment · Reviewer_35ZH · 2023-11-22
> > > **Thanks to Your Response**
> > >
> > > Thanks to your detailed response. Most of my concerns have been solved and I have updated my score.
> > >
> > > Good Luck!

---

> > > > ### Author Response · Authors · 2023-11-22
> > > > **Thank you!**
> > > >
> > > > We would like to express our sincere gratitude for your invaluable suggestions and appreciation throughout the review process. We thoroughly enjoyed communicating with you and greatly appreciate your time!

---

### Official Review · Reviewer_GMmw · 2023-10-30

**Soundness:** 4 excellent
**Presentation:** 4 excellent
**Contribution:** 4 excellent
**Rating:** 8
**Confidence:** 3

**Summary:**

This paper proposes `DrS` (Dense reward learning from Stages), an approach for learning reusable dense rewards for multi-stage tasks, effectively reducing human efforts in reward engineering. By breaking down the dask into stages, this method learns dense reward from sparse ones using human demonstrations, where the focus is to learn reusable representations that can potentially by used on unseen tasks with similar structure. An example of structure is illustrated using the `Open Cabinet Door` which can be naturally divided into stages, such as approaching the handle, grasping and pulling, and then releasing it. Recognizing which stage an agent is in can be done using simple binary indicators. By applying these indicators, this work cultivates a dense reward for every stage.

Experiments are `PickNPlace`, `TurnFaucet`, `OpenCabinetDoor`.

Baselines are `dense reward`,  `VICE-RAQ`, `ORIL`

**Strengths:**

- The goal of this work is deriving a dense reward function from an array of training tasks to be repurposed for new, unseen tasks.
- The notion of capturing representations for a `task family` is important for enabling RL agents to learn multi-purpose policies.
-  Operating on the understanding that tasks can be broken down into several segments is also logical.
- I like this paper because I think it's important to move away from engineered dense rewards, to more tangible methodologies for learning rewards, specially in stage-drived manner, using demonstrations.

**Weaknesses:**

- Some ablation study of the robustness of this method against bad demonstrations (i.e. suboptimal, noisy etc) could be nice.

**Questions:**

- I did not understand what Figure 3 is presenting and there is no reference in the paper. Does this plot mean that `DrS` is learning a sigmoid liked function of the sparse reward?
- In the limitation section, the authors talk about the usage of language models such as ChatGPT and I think some discussion can be built around methods such as say-can[1]. I think the goal of both frameworks are aligned.





[1] Do As I Can, Not As I Say: Grounding Language in Robotic Affordances -  https://say-can.github.io/

---

> ### Author Response · Authors · 2023-11-22
> **Author Response**
>
> We express our gratitude for your thorough review and favorable rating. It is heartening to know you deem **our method important and logical**. We sincerely appreciate your efforts in helping us improve the quality of our work! Below, we address the concerns you raised.
>
> ---
>
> > Some ablation study of the robustness of this method against bad demonstrations (i.e. suboptimal, noisy, etc) could be nice.
>
>
> In response to your suggestion, we have conducted additional experiments to assess the robustness of our method against noisy demonstrations. These experiments involved training our learned dense rewards with noisy demonstrations, followed by reusing them to train RL agents from scratch. The results are shown in Fig. 19 of Appendix I.2 in the updated version of our paper.
>
> The results demonstrate that the use of noisy demonstrations during the reward learning phase **does not substantially affect the quality of the learned reward**. This outcome aligns with our expectations, considering that the discriminator's positive training data is **dominated by success trajectories collected online**. By the end of the reward learning phase, there are hundreds of thousands of these online success trajectories, in contrast to only a few hundred success trajectories from demonstrations. **Therefore, the initial quality of the demonstrations becomes less critical.**
>
>
> Implementation Details:
>
> As outlined in Appendix A, our demonstrations were collected by trained RL agents. To introduce noise, we added Gaussian noise during the rollout, but only retained those success trajectories. This resulted in demonstrations that were successful, but characterized by more unstable motions.
>
> ---
>
>
> > I did not understand what Figure 3 is presenting and there is no reference in the paper. Does this plot mean that DrS is learning a sigmoid liked function of the sparse reward?
>
> Figure 3 is referenced in our paper, specifically in the second-to-last line of Sec. 4.4. The purpose of this figure is to illustrate the contrast between the reward structure learned by our method and a semi-sparse reward. Our learned reward offers smooth reward signals to the RL agents, in contrast to the semi-sparse reward, which provides uniform rewards within each stage.
>
> While DrS does look like a “sigmoid-like” function in a broad sense, a more precise description would be that it learns a reward curve within each stage that resembles a **tanh** function, as shown in Equation 4.
>
> ---
>
> > In the limitation section, the authors talk about the usage of language models such as ChatGPT and I think some discussion can be built around methods such as say-can[1]. I think the goal of both frameworks are aligned.
>
>
> Thank you for suggesting the relevant study. Yes, SayCan is a great paper and we **have updated our manuscript** to include a citation in the related works section.
>
> SayCan employs Large Language Models (LLMs) to break down tasks into smaller sub-tasks, each of which is then addressed using low-level skills. This concept of task decomposition into stages or sub-tasks is indeed akin to our approach. Furthermore, ideas similar to task decomposition are also explored in Hierarchical Reinforcement Learning approaches [1,2,3] and skill chaining methods [4,5,6].
>
> However, a key distinction of our work lies in our approach to utilizing stage structures. While the aforementioned methods apply stage structures primarily **within the policy space**, our work takes **an orthogonal direction by focusing on the design of rewards that incorporate stage structures**. This unique perspective on task decomposition through reward design is an important attribute of our work.
>
>
> [1] Kevin Frans, Jonathan Ho, Xi Chen, Pieter Abbeel, and John Schulman. Meta learning shared hierarchies. In International Conference on Learning Representations, 2018.
>
> [2] Ofir Nachum, Shixiang Shane Gu, Honglak Lee, and Sergey Levine. Data-efficient hierarchical reinforcement learning. Advances in Neural Information Processing Systems, 31:3303–3313, 2018.
>
> [3] Andrew Levy, George Konidaris, Robert Platt, and Kate Saenko. Learning multi-level hierarchies with hindsight. In International Conference on Learning Representations, 2018.
>
> [4] Youngwoon Lee, Joseph J Lim, Anima Anandkumar, and Yuke Zhu. Adversarial skill chaining for long-horizon robot manipulation via terminal state regularization. arXiv preprint arXiv:2111.07999, 2021.
>
> [5] Jiayuan Gu, Devendra Singh Chaplot, Hao Su, and Jitendra Malik. Multi-skill mobile manipulation for object rearrangement. arXiv preprint arXiv:2209.02778, 2022.
>
> [6] Youngwoon Lee, Shao-Hua Sun, Sriram Somasundaram, Edward S Hu, and Joseph J Lim. Composing complex skills by learning transition policies. In International Conference on Learning Representations, 2019.
>
> ---
>
> Hope the above clarifications and responses address your concerns. Thanks so much for your time and effort! Have a great Thanksgiving holiday :)

---

### Official Review · Reviewer_oxbF · 2023-10-31

**Soundness:** 2 fair
**Presentation:** 2 fair
**Contribution:** 1 poor
**Rating:** 3
**Confidence:** 4

**Summary:**

This paper proposes a method for learning reusable dense rewards for multi-stage tasks using binary stage indicators. The idea is to train a discriminator that distinguishes successful trajectories from unsuccessful ones for each stage, and then use the discriminator output as dense reward. The authors demonstrate that the learned reward functions generalize to thousands of tasks across three Maniskill domains, resulting in policies that perform nearly as well as those trained from human-designed rewards. This is a step towards automating reward design, which is a long-standing problem in RL.

**Strengths:**

- This paper makes a contribution towards automating reward design, which is of paramount importance in the field of RL. Having access to dense rewards takes the burden off of exploration, which in turn reduces the number of samples required to solve a task.
- The method is a niche application of contrastive discriminator learning, which is well-established in the literature.

**Weaknesses:**

- The method requires success and failure trajectories for each stage in the training data, which can be expensive to collect.
- The scope of the method is limited to a family of tasks that can be divided into stages. This prevents it from being applied to other tasks such as locomotion. It also means the method is less general compared to LLM-based rewards with external knowledge [1, 2].
- Similarly, the need for stage indicators prevents the method from scaling to real-world problems, which would arguably benefit more from automated reward design than simulated domains.
- The experiments do not demonstrate new skills that are unachievable by human-designed rewards.

References:
1. Tianbao Xie, Siheng Zhao, Chen Henry Wu, Yitao Liu, Qian Luo, Victor Zhong, Yanchao Yang, and Tao Yu. Text2reward: Automated dense reward function generation for reinforcement learning. arXiv preprint, 2023.
2. Yecheng Jason Ma, William Liang, Guanzhi Wang, De-An Huang, Osbert Bastani, Dinesh Jayaraman, Yuke Zhu, Linxi “Jim” Fan, Anima Anandkumar. Eureka: Human-Level Reward Design via Coding Large Language Models. arXiv preprint, 2023.

**Questions:**

- Does the method work with pixel observations?
- If the method only works with states, does generalization come as a result of state abstraction or the method itself?
- Do the learned dense reward functions resemble hand-designed rewards?

---

> ### Author Response · Authors · 2023-11-20
> **Author Response (1/3)**
>
> Thank you for your thoughtful and constructive feedback on our paper! We are delighted to learn that you found **we make a contribution towards automating reward design**. Below, we address the concerns you raised.
>
> ---
>
> > The method requires success and failure trajectories for each stage in the training data, which can be expensive to collect.
>
> Collecting trajectories is required in most reward learning methods, both online reward learning methods (e.g., traditional IRL [1,2,3], AIL [4, 5, 6], and other approaches [7, 8, 9]) and offline reward learning methods (e.g., [10, 11,12]). **Therefore, our method is not more expensive than these existing methods**. Moreover, the process of training RL agents inherently involves collecting trajectories.
>
> If the reviewer means labeling the trajectories with stage information is expensive, let’s discuss from the following two perspectives.
>
> If the reviewer's concern is about the cost of labeling trajectories with stage information, we can discuss this from two angles:
>
> 1. In Simulation: Assigning stage indicators within a simulator is simple, and often requires minimal code. For example, the stage indicators for the “Open Cabinet Door” tasks were implemented with just *three lines of code*. Further details can be found in Appendix B.
>
> 2. In Real World: Implementing stage indicators is also possible in real-world settings. A more detailed discussion on this is provided in the subsequent response which is marked as `[INCLUDING DISCUSSIONS FOR THE FIRST QUESTION]`.
>
>
> [1] Ng, Andrew Y., and Stuart Russell. "Algorithms for inverse reinforcement learning." Icml. Vol. 1. 2000.
>
> [2] Abbeel, Pieter, and Andrew Y. Ng. "Apprenticeship learning via inverse reinforcement learning." Proceedings of the twenty-first international conference on Machine learning. 2004.
>
> [3] Ziebart, Brian D., et al. "Maximum entropy inverse reinforcement learning." Aaai. Vol. 8. 2008.
>
> [4] Ho, Jonathan, and Stefano Ermon. "Generative adversarial imitation learning." Advances in neural information processing systems 29 (2016).
>
> [5] Fu, Justin, Katie Luo, and Sergey Levine. "Learning robust rewards with adversarial inverse reinforcement learning." arXiv preprint arXiv:1710.11248 (2017).
>
> [6] Kostrikov, Ilya, et al. "Discriminator-actor-critic: Addressing sample inefficiency and reward bias in adversarial imitation learning." arXiv preprint arXiv:1809.02925 (2018).
>
> [7] Singh, Avi, et al. "End-to-end robotic reinforcement learning without reward engineering." arXiv preprint arXiv:1904.07854 (2019).
>
> [8] Wu, Zheng, et al. "Learning dense rewards for contact-rich manipulation tasks." 2021 IEEE International Conference on Robotics and Automation (ICRA). IEEE, 2021.
>
> [9] Memarian, Farzan, et al. "Self-supervised online reward shaping in sparse-reward environments." 2021 IEEE/RSJ International Conference on Intelligent Robots and Systems (IROS). IEEE, 2021.
>
> [10] Smith, Laura, et al. "Avid: Learning multi-stage tasks via pixel-level translation of human videos." arXiv preprint arXiv:1912.04443 (2019).
>
> [11] Kalashnikov, Dmitry, et al. "Mt-opt: Continuous multi-task robotic reinforcement learning at scale." arXiv preprint arXiv:2104.08212 (2021).
>
> [12] Zolna, Konrad, et al. "Offline learning from demonstrations and unlabeled experience." arXiv preprint arXiv:2011.13885 (2020).
>
>
> ---
>
> > The scope of the method is limited to a family of tasks that can be divided into stages. This prevents it from being applied to other tasks such as locomotion.
>
> Actually, our method can applied to the tasks without stages. While our reward learning approach appears to require dividing tasks into stages, it is mainly because the tasks we used in this paper are **long-horizon**. For tasks where specifying stages is challenging(e.g., locomotion), let’s discuss them in two cases:
>
>
>
> 1. Short-Horizon Tasks: Our method (one-stage version) effectively handles short-horizon tasks without requiring stage information. For example, our approach can learn reusable rewards for HalfChettah (a representative locomotion task in MuJoCo). See **Appendix G.2 (in the updated PDF)** for details.
>
> 2. Long-Horizon Tasks: When a task is long-horizon and its stages are not easily definable, it presents a significant challenge. Even for human experts, designing a dense reward for such tasks is very challenging. Take the [Ant Maze](https://robotics.farama.org/envs/maze/ant_maze/) task as an example, which is long-horizon and hard to specify stages. The [default dense reward designed by humans](https://github.com/Farama-Foundation/Gymnasium-Robotics/blob/main/gymnasium_robotics/envs/maze/ant_maze_v4.py#L154) is merely the negative L2 distance, which is very bad since it ignores the walls in the maze. Therefore, we leave these tasks for future work.

---

> ### Author Response · Authors · 2023-11-20
> **Author Response (2/3)**
>
> > It also means the method is less general compared to LLM-based rewards with external knowledge [1, 2].
>
> Text2Reward [1] and Eureka [2] are both great papers and we **will cite them** in the revised manuscript following the conclusion of the discussion phase. (In fact, Text2Reward has already been cited in our initial submission.)
>
> Before delving into a comparison with these works, we would like to point out that they are concurrent with our research (both being submissions to ICLR 2024). Text2Reward was released a week before the ICLR submission deadline, and we have already included a comparison with it in Appendix C of our submitted version. Eureka was released after this deadline, making it impossible for us to include a comparison in our initial submission. Although comparing with concurrent studies might seem somewhat unfair to us, we still provide a discussion below.
>
> *Our reward learning approach is not directly comparable to those LLM-based rewards due to the differences in problem setups*. While LLM-based rewards appear more generally applicable to a range of tasks, they impose more specific requirements on the setup. The key differences are as follows:
>
> 1. Access to Source Code: LLM-based rewards (both Text2Reward and Eureka) assume access to the source code of the ask. Our method, in contrast, relies solely on success signals or stage indicators, without the need for task code.
>
> 2. Instruction-Reward Pairs: LLM-based rewards require a pool of instruction-reward code pairs (as in Text2Reward) or other instructional materials for coding a reward (as in Eureka). Our method, alternatively, utilizes stage indicators.
>
> 3. Environment Interaction: Both LLM-based rewards and our approach require interaction with the environment. However, our method learns the reward in a single run, whereas LLM-based rewards (both Text2Reward and Eureka) require multiple rounds of RL training for reward improvement.
>
> 4. Evaluation Focus: Our approach places greater emphasis on evaluating learned rewards on unseen test tasks. In contrast, LLM-based rewards (both Text2Reward and Eureka) directly apply rewards to test tasks for iterative feedback and improvement.
>
>
> In conclusion, while both LLM-based rewards and our approach aim to automate reward design, they cater to distinctly different scenarios.
>
> ---
>
> > Similarly, the need for stage indicators prevents the method from scaling to real-world problems, which would arguably benefit more from automated reward design than simulated domains.
>
> `[INCLUDING DISCUSSIONS FOR THE FIRST QUESTION]`
>
> As we discussed in Sec. 3 in our paper, the stage indicators are only required during RL training, but **not required when deploying the policy to the real world**. Training RL agents directly in the real world is often impractical due to cost and safety issues. Instead, a more common practice is to train the agent in simulators and then transfer/deploy it to the real world.
>
> If one really wants to train RL agents directly in the real world, computing stage information still **highly likely requires less effort than computing a full dense reward (from human or LLM)**. In the real world, the stage information can be obtained by many existing techniques. Taking “Pick-and-Place” as an example, the “object is grasped” indicator can be obtained by tactile sensors, and the “object is placed” indicator can be obtained by robot proprioception and forward kinematics or visual detection/tracking techniques.
>
> ---
>
> > The experiments do not demonstrate new skills that are unachievable by human-designed rewards.
>
>
> The primary goal of our paper is to reduce the human effort involved in reward engineering, rather than solving the tasks that are unachievable with human-engineered rewards.
>
> Moreover, we posit that if a task can be effectively addressed by automatically generated or learned rewards (such as LLM-based rewards or our method), it is likely that human experts could also devise a comparable reward, given sufficient time and effort. Thus, claiming that certain rewards are beyond the reach of human experts seems somewhat unrealistic.

---

> > ### Author Response · Authors · 2023-11-20
> > **Author Response (3/3)**
> >
> > > Does the method work with pixel observations? If the method only works with states, does generalization come as a result of state abstraction or the method itself?
> >
> > Our method does work with high-dimensional visual inputs. Rather than using pixel observations, we conducted experiments using point cloud inputs, as they better capture the geometric information of objects. The results are detailed in Appendix E.2.
> >
> > The results demonstrate that rewards derived from point cloud inputs are comparable to those obtained with state inputs. This indicates that our method is fully compatible with high-dimensional visual inputs. However, the techniques for visual inputs are somewhat orthogonal to our primary focus, which is reward learning. Given that experiments involving visual inputs require considerably more time, we primarily conducted our experiments using state inputs.
> >
> > ---
> >
> > > Do the learned dense reward functions resemble hand-designed rewards?
> >
> > The similarity between learned dense rewards and hand-designed rewards largely depends on how the “hand-designed” rewards look like. For example, the human-engineered reward for the “Open Cabinet Door” tasks is very complicated, and it involves over 100 lines of code, 10 candidate terms, and tons of “magic” parameters. Given that so many parameters are manually defined in such human-engineered rewards, it is unlikely that the learned rewards would mirror them exactly — and that's not necessarily a disadvantage.
> >
> > A straightforward way to check whether the learned rewards align with human intentions is through visualization. However, visualizing rewards in high-dimensional spaces poses challenges. Therefore, we instead conduct our visualization in a 2D navigation environment. Figure 16 in Appendix G.1.3 visualizes the learned reward in this setting, with a detailed explanation provided in the figure caption. The visualization reveals that the learned reward effectively directs the agent towards the goal, which perfectly aligns with human intentions.
> >
> >
> > ---
> >
> > Hope the above clarifications and responses address your concerns. If not, feel free to let us know, and **we are more than happy to answer additional questions**. If you feel that our rebuttal has addressed your concerns, we would be grateful if you would consider **revising your score in response**. Thanks so much for your time!

---

> ### Comment · Reviewer_oxbF · 2023-11-20
>
> Thanks for addressing my questions. I believe the method of this paper is not quite novel. The learning of each stage discriminator is fundamentally GAIL with a looser assumption: GAIL doesn't query the environment for success signals at training time whereas DrS does. Hence GAIL has the converging discriminator problem: it has to get the negative trajectories somewhere, and that is the agent's current behavior policy. This means the comparison to GAIL isn't really apples to apples. However, there is merit in applying existing methods to new problem formulations and developing a strong practical method. My remaining concerns mainly revolve around the practicality of the method.
> - The multistage version of the method requires augmenting the state space with a stage indicator. This addition is not organic, because we need to modify the state space to be compatible with the reward function. So even if we perform sim2real transfer, we still need to have the stage indicator in real to ensure compatibility with the reward function. I'm not sure how we can get around that. Is it ok to discard the stage indicator dimensions when deploying the policy in the real world? Does the policy not rely on any information from the stage indicator?
> - The most accessible modality of real-world RL is pixels. Point clouds require depth information and can be slow to run. I wonder if there is a reason for the omission of experiments from pixels, given it is strictly easier to run than from point clouds. If this is because of negative results, I encourage you to include it as there is value to understanding why the performance is not so good.

---

> > ### Author Response · Authors · 2023-11-20
> > **2nd Round of Author Response**
> >
> > Thank you for taking the time to read our rebuttal and responding in time! We are delighted to learn that you recognize **our merit in applying existing methods to new problem formulations and developing a strong practical method**.
> >
> > Below, we address the additional concerns you raised in the response. ( **We assume all other issues raised in the initial review have been addressed, but please inform us if any remain unresolved.** )
> >
> > ---
> >
> > >  I believe the method of this paper is not quite novel. The learning of each stage discriminator is fundamentally GAIL with a looser assumption…
> >
> > While the assessment of novelty in academic work can often be subjective, we respectfully contest the viewpoint that “the learning of each stage discriminator is fundamentally GAIL with a looser assumption.”
> >
> > Generative Adversarial Imitation Learning (GAIL) is characterized primarily by its **adversarial** nature. In GAIL, the discriminator (reward) aims to differentiate between expert and agent trajectories, while the generator (policy) tries to **fool the discriminator** by generating trajectories akin to those of the expert. In contrast, our approach **does not adhere to this adversarial paradigm**. In our approach, the discriminator distinguishes between successful and unsuccessful trajectories, but the generator (policy) **is not tasked with fooling the discriminator**. Rather, it seeks to maximize the expected return through an RL objective. **This shift away from adversarial learning is a significant departure from GAIL**. *We believe it is somewhat unfair to label our approach as fundamentally GAIL, considering our divergence from its core adversarial feature.* Additionally, our method also introduces a novel mechanism for leveraging multi-stage information. We totally respect the reviewer's perspective, but it is crucial to highlight these fundamental differences.
> >
> > ---
> >
> > >  … Is it ok to discard the stage indicator dimensions when deploying the policy in the real world? Does the policy not rely on any information from the stage indicator?
> >
> > Our method does not need to modify the state space to be compatible with the reward function. Both the discriminator and the policy do not take stage indicators as the inputs. The role of these indicators is to select discriminators and to assign trajectories to specific buffers. As a result, there is no need to consider any stage information when deploying the policy in the real world.
> >
> > ---
> >
> > > Point clouds require depth information and can be slow to run. I wonder if there is a reason for the omission of experiments from pixels, given it is strictly easier to run than from point clouds…
> >
> > Actually, one important reason we opted for point clouds is that it is *much faster* compared to the pixel-based variant. Point clouds allow us to easily filter out the points on the ground, resulting in a **significantly smaller** visual input than the original RGB images. With such a small point cloud, we can just use a very lightweight PointNet to extract features. In contrast, the CNN used to process the RGB images would be much heavier. Therefore, running the experiments with point clouds is a more viable choice based on practical considerations of experiment time and computational resource constraints.
> >
> > ---
> >
> > Once again, we extend our sincere gratitude for your time and thoughtful consideration. If you believe that our rebuttal has successfully addressed your concerns, we would be deeply appreciative if you would consider **revising your score accordingly**. Thank you very much for your attention and support!

---

> ### Comment · Reviewer_oxbF · 2023-11-20
>
> - I raised my concern about stage indicators when reading your discussion with reviewer 35ZH. There, you mentioned that the reward of a state never changes because the same state in different stages is disambiguated by a stage indicator. Hence I thought you augmented the state space with the stage indicator, which turned out to be not true. But this raises another question: how does your policy distinguish the same state from different stages, when the policy doesn't have access to the stage indicator? Are you using a non-Markovian policy?
> - Thanks for the clarification about point clouds. From my experience, dense point clouds are slower than pixels, but perhaps the filtering helped. Still, this doesn't justify the omission of pixel experiments, as it should not be that much more expensive to run. Is there a reason for the omission besides the heavy time consumption?

---

> ### Author Response · Authors · 2023-11-21
> **3rd Round of Author Response**
>
> Thank you for the prompt reply! **We assume all other issues raised in the previous review have been addressed, but please inform us if any remain unresolved.**
>
> > … how does your policy distinguish the same state from different stages, when the policy doesn't have access to the stage indicator? Are you using a non-Markovian policy?
>
> We would like to kindly request further elaboration from the reviewer on the statement regarding the ability to “distinguish the same state from different stages.” In our setup, each state is uniquely associated with a specific stage, as illustrated in Fig. 1, meaning that **the same state does not occur in different stages**. For example, if the robot has grasped the handle but has not yet opened the door, then this state is unambiguously in stage 2. Additionally, the policy's role is solely to generate an action; it does not need to identify the stage of a state. If there is a need for a more detailed explanation or if any aspect of our approach remains unclear, please let us know. We are more than willing to provide any additional clarification required.
>
> ---
>
> > … Is there a reason for the omission besides the heavy time consumption?
>
> As mentioned in our initial response, we chose point clouds because they better capture the geometric information of objects. Additionally, experiments utilizing point clouds are less time-consuming and require fewer computational resources, making them a more practical choice.
>
> It is important to note that techniques involving visual inputs are somewhat orthogonal to our research focus on reward learning. However, **if the reviewer deems the absence of results on pixel observations as a major reason for rejection**, we could run these experiments immediately. While we cannot guarantee completion within the remaining rebuttal period due to time constraints, we assure you that we will make every effort to do so, **if it is considered essential**.
>
> ---
>
> Thank you again for your time and effort! If you feel that our rebuttal has addressed your concerns, we would be grateful if you would consider **revising your score in response**.

---

> > ### Comment · Reviewer_oxbF · 2023-11-21
> >
> > Thanks for answering my question about the policy. It is indeed possible to avoid ambiguous states with a carefully designed state space. However, given the discussion so far, I have plenty of reasons to believe that **the claimed generalization capabilities are a result of state abstraction**, which has little to do with the method. Here are the reasons why:
> >
> > 1. It is evident that state abstraction enables generalization of learned reward functions. Consider the pick and place task. A possible reward function is distance from endeffector to object center of mass (COM) for the first stage, and distance from object COM to the target position for the second stage. If the state consists of the endeffector position and the object COM, then this simple reward function can generalize to thousands of objects with different shapes. Similarly, if we provide the handle position and cabinet hinge angle in the states for the cabinet task, or the handle position and knob angle for the faucet task, then a simple human engineered reward could also generalize to different instantiations.
> >
> > 2. As stated in Appendix E.2, the point cloud experiments *also provide low dimensional states as inputs to the reward function*. The reward function can theoretically ignore the point cloud and still be able to generalize. Hence, there is no evidence that the learned reward functions can generalize from other modalities than states.
> >
> > 3.  There is no obvious reason to omit the pixel experiments. I suspect this is because pixel-based reward functions don't generalize very well. The addition of point cloud already hurts performance, so higher-dimensional pixels might affect the performance even more.
> >
> > The majority of the paper emphasizes its generalization capabilities, but for the aforementioned reasons, I believe these capabilities are a result of state abstraction, not the proposed method. I cannot say if the absence of certain results is critical for rejection, but I suggest making the following revisions to the paper:
> > 1. Provide experiments with pixel inputs. I requested this in my initial review, so time is not an excuse.
> > 2. Provide experiments of sim2real transfer, i.e. training policies in sim with learned rewards and transfering them to the real world. This is another claim made in the paper but not validated.
> > 3. Revise the paper to tone down the emphasis on generalization, unless there is evidence suggesting that generalization is indeed a contribution.

---

> ### Author Response · Authors · 2023-11-21
> **4th Round of Author Response**
>
> Thank you for your prompt response!
>
> We would like to address what appears to be a **significant misunderstanding** regarding our paper. The primary criticisms from the reviewer seem to focus on an assumed emphasis on reward generalizability in our work. However, **we did not claim reward generalizability as a contribution**. In fact, terms like **“generalization,” “generalizability,” or “generalize” are only mentioned once in our main text, and the only occurrence refers to policy, not reward**. Furthermore, **it did not seem to cause confusion among other reviewers, as they did not mention anything about generalization in their reviews**.  Therefore, *we believe it is both unfair and unfounded to criticize our paper based on a claim we never made*.
>
> Instead of generalizability, we actually claimed that our rewards are **reusable**.  A reward is considered reusable if, once learned on training task A, it can train an RL agent from scratch at least on the same task A. Hence, **reusability is a necessary but not a sufficient condition for generalizability**. We apologize if the use of terms about “reusability” has caused any confusion. Our intention was to minimize misunderstanding by **deliberately avoiding terms related to “generalization”**.
>
> The rationale behind prioritizing reusability over generalizability stems from the limitations of existing AIL methods in producing reusable rewards. For example, a reward from GAIL, when reused on the same task it was learned from, fails to train an RL agent from scratch. This issue is not about a generalization gap; it is about the inherent non-reusability of GAIL rewards. Our objective is to address this issue and learn a reward that is reusable. Because reusing a reward on training tasks is not very interesting (as we already have the trained policies for these tasks), we chose to evaluate our rewards' reusability on unseen test tasks. Again, this evaluation was aimed at assessing the reusability of our learned rewards, not their generalizability.
>
> Furthermore, the reviewer also criticizes using low-dimensional states as the input to reward. However, **this is the only fair way to compare with human-engineered reward** since human-engineered reward requires oracle state information, and **it even uses more state information than our learned reward.** While we do acknowledge the role of state abstraction in reward generalization, we did not aim to achieve better generalization than human-engineered rewards by any means (because we feel it is nearly impossible).
>
> Finally, we want to re-iterate the primary focus of our paper. We aim to **reduce the human effort** involved in reward engineering by **learning reusable dense rewards**. We do not claim generalization as our contribution, but it can be a natural outcome of our learned reward.
>
>
>
> We hope this clarification addresses the reviewer's concerns. **We genuinely appreciate the time and effort invested in reviewing our paper, and we believe that not all the reviewers are as patient as you.** We are more than willing to provide further clarification to avoid any major misunderstandings. Thank you once again for your valuable time.

---

### Official Review · Reviewer_ck68 · 2023-11-01

**Soundness:** 4 excellent
**Presentation:** 4 excellent
**Contribution:** 3 good
**Rating:** 8
**Confidence:** 3

**Summary:**

The paper presents a method, Dense reward learning from Stages (DrS), for learning reusable dense reward functions using only sparse task reward signals and task-family-specific “stage indicators”. DrS learns to classify the stage reached by each trajectory and adds the discriminator score to the index of the stage reached, leading to a reward function that increases both across stages and within progress along a stage. Prior adversarial imitation learning approaches to reward learning do not lead to reward functions that can be reused because the discriminator cannot distinguish policy trajectories from demonstrations at convergence. Experiments show that DrS-learned rewards can be reused for test tasks and can even compete with human-engineered rewards in some cases, despite requiring much less manual engineering (just the stage indicator function).

**Strengths:**

* As far as I am aware, the overall design of the algorithm (exploiting stage indicators) as well as the form of the learned reward function are novel.
* The method significantly reduces the amount of engineering required to learn reward functions when the task can be broken down into identifiable stages.
* DrS handily outperforms several reasonable baselines and competes with the hand-engineered reward in some cases.
* In addition to the ablation study in the body of the paper, the appendix includes extensive additional experiments, such as varying the number of training tasks, input modality, dynamics, action space, robot morphology, and form of the reward function. The authors also verify the claim that GAIL-generated rewards are not reusable.
* The paper is clear, easy to read, and well-motivated.

**Weaknesses:**

* The paper states that demonstrations are optional, but it sounds like they were used in all experiments. I imagine that sample efficiency would deteriorate substantially if no demonstrations are provided and the first stage cannot be solved easily by random exploration, or more generally if any stage cannot be easily solved by a noisy version of a policy that solves the previous stage.
* It is not clear that all sparse-reward tasks can be broken up into stages, and as shown in the ablation study, the method struggles when there is only one stage. So DrS is not universally applicable to sparse-reward tasks.

**Questions:**

Did you try running DrS without providing demonstrations? Or varying the number of demonstrations?

---

> ### Author Response · Authors · 2023-11-22
> **Author Response**
>
> We are deeply grateful for your review and positive evaluation! It is encouraging to receive recognition for **our method's novelty, strong performance, extensive experiments, and clear presentation**. We sincerely appreciate your efforts in helping us improve the quality of our work! Below, we address the concerns you raised.
>
> ---
>
> > The paper states that demonstrations are optional, but it sounds like they were used in all experiments. I imagine that sample efficiency would deteriorate substantially if no demonstrations are provided …
>
> > Did you try running DrS without providing demonstrations? Or varying the number of demonstrations?
>
>
> Before addressing your questions, we would like to clarify that our approach involves two phases: the **Reward Learning Phase** and the **Reward Reuse Phase**. Demonstrations are an optional component in the Reward Learning Phase, but are not utilized in the Reward Reuse Phase.
>
>
> Below is the response to your questions:
>
> Yes, we did use demonstrations in the **reward learning phases** of all experiments because the demonstrations *could help to kick off the training of rewards*. And your imagination is correct - without demonstrations, *we can still learn the reward*; however, the training would be *slower* and have *a larger variance* because the agent must explore by itself to find the first sparse reward signal before starting the reward training. Therefore, we strongly recommend to utilize demonstrations when they are available.
>
> However, our reward reuse phase **will not be influenced by the demonstrations** after the reward is learned. This is because the positive training data of the discriminator is **dominated by the online collected success trajectories**. Specifically, when the training converges, there are hundreds of thousands of online success trajectories collected by the agent, whereas only a few hundred success trajectories are from the demonstrations.
>
> The relevant results are shown in Fig. 18 of Appendix I.1 in our updated PDF. From the figure, it is evident that the use of demonstrations in the reward learning phase **does not significantly impact the quality of the learned reward**.
>
> In summary, demonstrations are not necessary for our approach, and they **do not considerably affect the quality of the learned reward**. However, we recommend utilizing a few demonstrations in practice because it saves a significant amount of time during reward learning.
>
> *Implementation Details:*
>
> If demonstrations are unavailable, the discriminators may lack positive training data in the early stages of training. Therefore, we initiate training the discriminator $i$ when there is at least one trajectory in buffers $i+1$ to $n$, i.e., we have a trajectory that goes beyond stage $i$. When the discriminator $i$ is not yet trained, we use a sparse reward for stage $i$.
>
> ---
>
> > It is not clear that all sparse-reward tasks can be broken up into stages, and as shown in the ablation study, the method struggles when there is only one stage. So DrS is not universally applicable to sparse-reward tasks.
>
>
> That is a valid observation! While we acknowledge that not all tasks can be easily segmented into stages, our reward learning approach requires the stage information mainly because the tasks are **long-horizon**. For the tasks that are hard to break up into stages (e.g., locomotion tasks), let’s discuss them in two cases:
>
> 1. Short-Horizon Tasks: Our method (one-stage version) effectively handles short-horizon tasks without requiring stage information. For example, our approach can learn reusable rewards for HalfChettah (a representative locomotion task in MuJoCo). See Appendix G.2 for details.
>
> 2. Long-Horizon Tasks: When a task is long-horizon and its stages are not easily definable, it presents a significant challenge. Even for human experts, designing a dense reward for such tasks is very challenging. Take the [Ant Maze](https://robotics.farama.org/envs/maze/ant_maze/) task as an example, which is long-horizon and hard to specify stages. The [default dense reward designed by humans](https://github.com/Farama-Foundation/Gymnasium-Robotics/blob/main/gymnasium_robotics/envs/maze/ant_maze_v4.py#L154) is merely the negative L2 distance, which is very bad since it ignores the walls in the maze. Therefore, we leave these tasks for future work.
>
>
> ---
>
> Hope the above clarifications and responses address your concerns. Thanks so much for your time and effort! Have a great Thanksgiving holiday :)

---

> > ### Comment · Reviewer_ck68 · 2023-11-22
> > **Response to authors**
> >
> > Thank you for conducting additional experiments in response to my questions! I would say that my concerns have been addressed and I still recommend acceptance.

---

### Comment · Area_Chair_XpeB · 2023-11-23
**Author-Reviewer discussion period ending *very* soon**

Thank you to the reviewers for responding and participating in the discussion phase. This phase is now coming to a close, so please post any final comments you have.
Thank you for your service!

---

### Author Response · Authors · 2023-11-23
**Clarification from authors**

Dear AC and Reviewers,

We are immensely grateful for your time, insightful feedback, and valuable suggestions throughout this review process. As the author-reviewer discussion period draws to a close and the reviewer-AC discussion phase will start soon, we wish to provide some clarifications regarding our paper to **prevent any potential misunderstandings** in your upcoming discussions or virtual meetings.

The primary focus of our paper is to **reduce the human effort** involved in reward engineering by **learning reusable dense rewards**. Specifically, a reward is considered reusable if, once learned on training task A, it can train an RL agent from scratch at least on the same task A. Hence, reusability is a necessary but not sufficient condition for generalizability. Because reusing a reward on training tasks is not very interesting (as we already have the trained policies for these tasks), we chose to evaluate our rewards' reusability on unseen test tasks. This evaluation was aimed at assessing the reusability of our learned rewards, not their generalizability. It is important to note that **we never claim generalization as our contribution**, but *it can be a natural outcome* of our learned reward.

We acknowledge that **most reviewers have accurately understood and reflected the main objectives and contributions of our paper in their reviews**. Nevertheless, **some critiques seem to be predicated on an assumed emphasis on reward generalizability, which is not our claim**. In the initial submission version of our manuscript, we **deliberately avoided using terms related to “generalization”** to minimize misunderstanding (terms like “generalization,” “generalizability,” or “generalize” are only mentioned once in our main text, and the only occurrence refers to policy, not reward).

We **deeply respect** the time and effort invested by **every reviewer** and understand that misunderstandings can occasionally arise. With this clarification, we hope to ensure that the upcoming reviewer-AC discussion is grounded in a clear and accurate understanding of our work.

Once again, we extend our thanks to all the reviewers and the AC for your dedication and the time you have devoted to reviewing our paper!

Sincerely,

The Authors

---

### Meta-Review · Area_Chair_XpeB · 2023-12-05

**Metareview:**

This work introduces DrS (Dense reward learning from Stages), a method for autonomously learning dense rewards for multi-stage tasks without heavy human intervention. By utilizing task stage structures, DrS extracts high-quality dense rewards from sparse rewards and demonstrations. These learned rewards are applicable to new tasks, reducing the need for manual reward engineering. After a fruitful rebuttal period, 2 reviewers suggest accept (8), one review leans towards accept (6), and another argues (quite strongly) for reject (3). Other than reviewer oxbF, the paper is mostly positive, this AC has therefore looked in detail at the issues presented by oxbF. First of all, it was great to see such a timely discussion between review and author, so thank you both for your time. This AC believes that all concerns raised by oxbF have been addressed, except for the one related to generalisation. After reading the paper, this AC believes that the authors do not claim generalisation as a contribution, and therefore, I find that the majority of oxbF's concerns have been addressed. Moreover, generalisation was not highlighted as an issue by the other 3 reviewers.

**Justification For Why Not Higher Score:**

The paper had mostly positive reviews, but none were championing the strongly.

**Justification For Why Not Lower Score:**

N/A

---

### Decision · Program_Chairs · 2024-01-16

Accept (poster)